# SOM-CPC: Unsupervised Contrastive Learning with Self-Organizing Maps for Structured Representations of High-Rate Time Series

## Abstract

Continuous monitoring with an ever-increasing number of sensors has become ubiquitous across many application domains. Acquired data are typically high-dimensional and difficult to interpret, but they are also hypothesized to lie on a low-dimensional manifold. Dimensionality reduction techniques have, therefore, been sought for. Recently, expressive non-linear deep learning (DL) models have gained popularity over more conventional methods like Principle Component Analysis (PCA) and Self-Organizing Maps (SOMs). However, the resulting latent space of a DL model often remains difficult to interpret. In this work we propose SOM-CPC, a model that jointly optimizes Contrastive Predictive Coding and a SOM to find an organized 2D manifold, while preserving higher-dimensional information. We address a largely unexplored and challenging set of scenarios comprising high-rate time series, and show on both synthetic and real-life data (medical sleep data and audio recordings) that SOM-CPC outperforms both DL-based feature extraction, followed by PCA, K-means or a SOM, and strong deep-SOM baselines that jointly optimize a DL model and a SOM. SOM-CPC has great potential to expose latent patterns in high-rate data streams and may therefore contribute to a better understanding of many different processes and systems.

## 1 Introduction

The improvement and abundance of sensor technology has led to large amounts of high-dimensional, information-rich continuous data streams. However, gaining actionable insights from these data is challenging due to their low interpretability. The main objective of this study is, therefore, to develop an algorithm for acquiring a structured and interpretable representation of (high-rate) time series. We define such an interpretable representation as one that has the ability to be informative and to facilitate exploration of the underlying structure (Lipton, 2018).

According to the manifold hypothesis, high-dimensional real-world data lies on a low-dimensional manifold, comprising disentangled latent factors of variation. The area of unsupervised representation learning is concerned with models that learn this manifold from a set of training data, without the bias of human annotations. Dimensionality reduction techniques like Principle Component Analysis (PCA), possibly in combination with clustering methods like K-means clustering, have conventionally been used for this purpose. Acquiring an interpretable representation with PCA requires omitting many principle components in order to achieve an interpretable number of components. This, however, may discard important information that can not linearly be projected on these few dimensions. A Self-Organizing Map (Kohonen, 1990), on the other hand, is an extension of K-means clustering that creates a low-dimensional interpretable visualization, while still representing the data in multiple dimensions. However, SOMs typically act on features, which need to be selected heuristically and may, therefore, strongly depend on the use case and/or data modality.

Deep learning (DL) models have become popular alternatives for non-linear dimensionality reduction that can be applied directly on raw data. Such models have been combined with joint clustering objectives in the latent space (Xie et al., 2016; Yang et al., 2017; Madiraju, 2018; Lee & Schaar, 2020). These methods, however, do typically not create a (visually) interpretable representation, and sometimes make use of label information during training (Lee & Schaar, 2020). To enhance

interpretability, latent space representations of DL models are often visualized using a t-distributed stochastic neighbor embedding (t-SNE) (Hinton & Roweis, 2002). Albeit its frequent use, t-SNE does not allow a direct deployment on unseen data as it does not learn a reusable mapping between the multi-dimensional and the low-dimensional space.

To acquire visually interpretable data representations from raw data, without assuming that data must live in two or three dimensions only, non-linear DL encoders have been combined with SOMs (Ferles et al., 2018; Pesteie et al., 2018; Fortuin et al., 2019; Forest et al., 2019; Manduchi et al., 2021; Forest et al., 2021). In the resulting joint training strategy of these deep-SOM models, the SOM objective can be seen as a regularizer on the encoding procedure, as it promotes a cluster-friendly feature space. Most of these models have focused on autoencoders as feature extractors. However, similar to Mrabah et al. (2020), we hypothesize that their reconstruction objective may hamper the clustering or structured representation learning objective: while within-cluster similarities should remain preserved for latent clustering, reconstruction demands a preservation of all factors of similarity. Moreover, in the context of time series representation learning, other self-supervised models - that take the temporal nature of the data into account during training - might be more suitable.

Contrastive self-supervised learning approaches have quickly become popular thanks to their superior representation learning performance in many domains (see Le-Khac et al. (2020) for a review). While many of these models rely on data augmentations during training in order to construct pairs of similar data points, Contrastive Predictive Coding (CPC) (Oord et al., 2019) leverages the temporal dimension for this purposes, making it a natural choice for self-supervised representation learning of time series. In CPC, the temporal dimension not only serves as a pretext task, but simultaneously enforces latent smoothness over time. The contributions of this work are as follows:

- We propose a new model in the deep-SOM family: *SOM-CPC*, which is suitable for learning structured and interpretable 2D representations of (high-rate) time series by encoding subsequent data windows to a topologically ordered set of quantization vectors.
- Using regression and classification probing tasks, we show that SOM-CPC preserves more information in its 2D representation than CPC that is followed by PCA, and a linear classifier or K-means, or directly encoding CPC's latent space to two dimensions. SOM-CPC's joint optimization, moreover, facilitates a smooth temporal trajectory through 2D space.
- We show that SOM-CPC quantitatively and qualitatively outperforms deep-SOM models with a reconstruction objective in terms of both clustering and topological ordering. It, moreover, requires less auxiliary loss functions (and associated hyperparameter tuning) thanks to its natural tendency to incorporate temporal smoothness. Lastly, SOM-CPC's training behavior shows that the SOM clustering objective better aligns with the CPC objective than with a reconstruction loss.

## 2 PRELIMINARIES

### 2.1 KOHONEN SELF-ORGANIZING MAPS

Kohonen's Self-Organizing Map (SOM) (Kohonen, 1990) is an algorithm to find a visually interpretable topological data representation. It has been found useful to reveal intricate patterns and structure in a plethora of applications. The algorithm's output, the low-dimensional visualization, is often referred to as a SOM as well. We choose to use a use a 2D visualization to enhance interpretability.

We define a set of data points $\mathcal{Z}$, and quantized counterparts $q_\Phi(z) \in \Phi$ for $z \in \mathcal{Z}$. The set $\Phi : \{\phi_1, \ldots, \phi_k\}$ is a trainable quantization codebook containing $k$ vectors or prototypes $\phi_i \in \mathbb{R}^F, 1 \leq i \leq k$. The j[th] prototype $\phi_{i=j}^{(n)} = q_\Phi(z)$ is the 'winning vector' for data point $z$, at iteration $n$ of the training procedure. The learned codebook vectors are placed on a pre-defined 2D grid by assigning an xy-coordinate to each vector at initialization. Note that this creates a 2D representation, while each data point $z$ still lives in $\mathbb{R}^F$, with $F \gg 2$. This is conceptually different than the way in which PCA achieves dimensionality reduction to 2D, where all information in the 3[rd] and higher principle components is strictly omitted. During training of a SOM, each $\phi_i$ is updated as follows (Kohonen, 1990), with $z \in \mathcal{Z}$:

$$\phi_i^{(n+1)} = \phi_i^{(n)} + \eta^{(n)} \mathcal{S}_i(\phi_{i=j}^{(n)})(z - \phi_i^{(n)}), \quad (1)$$

where $\eta^{(n)}$ is a time-decreasing learning rate. Topological neighborhood structure is promoted via a neighbourhood kernel $\mathcal{S}$ that weighs nodes inversely proportional to their distance with the winning node. A Gaussian kernel is often used which weighs node $i$ according to:

$$\mathcal{S}_i\big(\phi_{i=j}^{(n)}\big) = \exp\Big(-\frac{d_{j,i}^{(n)}}{2(\sigma^{(n)})^2}\Big), \quad \text{with} \tag{2}$$

$$d_{j,i}^{(n)} = ||\mathcal{P}\{\phi_{i=j}^{(n)}\}, \mathcal{P}\{\phi_i^{(n)}\}||_2^2 \quad \text{and} \quad \sigma^{(n)} = \sigma^{(0)}\exp(-n/\lambda), \tag{3}$$

where $\mathcal{P}$ projects a codebook vector to its corresponding coordinate on the grid, $\sigma^{(0)}$ denotes the initial standard deviation, and $\lambda$ the decay factor. Setting $\lambda = -n_{\max}/\log(\sigma^{(n_{\max})}/\sigma^{(0)})$ sweeps $\sigma$ between $\sigma^{(0)}$ and $\sigma^{n_{\max}}$ in $n_{\max}$ steps. The dependence of $\mathcal{S}_i$ on the distance $d_{j,i}$, implies a weighing of 1 for the winning node (i.e. distance equals zero), and lower than 1 for neighbour nodes. Note that other neighbourhood structures have been proposed as well, for example using the four closest neighbours on the grid, which results in a kernel with a *plus*-shape (Fortuin et al., 2019).

## 2.2 DEEP-SOM MODELS

All deep-SOM research has focused on combining autoencoders (Ferles et al., 2018; Pesteie et al., 2018; Fortuin et al., 2019; Forest et al., 2019; Manduchi et al., 2021; Forest et al., 2021) with a SOM. These models can broadly be summarized as a vector-quantized (VQ) VAE (van den Oord et al., 2017), with a topological organization of the vectors in the quantization codebook: the SOM. The models are trained end-to-end using error backpropagation of both a reconstruction *task* loss $\mathcal{L}_{\text{task}}$ and a loss $\mathcal{L}_{\text{topo}}$ that encourages *topological* ordering in the SOM. In general, a deep-SOM training objective takes the following form:

$$\mathcal{L}_{\text{deep-SOM}} = \mathcal{L}_{\text{task}} + \alpha\mathcal{L}_{\text{topo}}, \quad (4) \quad \text{with} \quad \mathcal{L}_{\text{topo}}(\boldsymbol{z}^{(n)}) = \mathbb{E}_{\mathcal{Z}}\Big[\sum_{i=1}^{k}\mathcal{S}_i\big(\phi_{i=j}^{(n)}\big)||\boldsymbol{z}^{(n)} - \phi_i^{(n)}||_2^2\Big]. \quad (5)$$

Hyperparameter $\alpha$ controls the trade-off. The topological loss thus replaces the original update rule of the SOM algorithm (see eq. (1)). The features $\boldsymbol{z} \in \mathcal{Z}$ are jointly optimized, and thus also depend on $n$ now. To prevent clutter we will, however, omit the (n)-superscript in the following.

Fortuin et al. (2019) propose the SOM-VAE model. As opposed to VQ-VAE, SOM-VAE has two decoders, as it also decodes the continuous latents. Topological organization of the codebook vectors is enforced by using a plus-shaped neighbourhood kernel, which affects the codebook vectors of the direct neighbours of the winning node (i.e. up, down, left, and right on the grid). The encoder parameters are, however, unaffected by the quantization error of these neighbour nodes. To facilitate the latter, the topological loss was split in a *commitment* loss (committing the winning codebook vector to $\boldsymbol{z}$ and *vice versa*) and a *SOM* loss (pulling the codebook vectors of the neighbours to $\boldsymbol{z}$): $\mathcal{L}_{\text{topo}} = \mathcal{L}_{\text{commitment}} + \frac{\beta}{\alpha}\mathcal{L}_{\text{SOM}}$. Formally:

$$\mathcal{L}_{\text{commitment}} = \mathbb{E}_{\mathcal{Z}}\Big[||\boldsymbol{z} - \phi_i||_2^2\Big], \quad (6) \quad \text{and} \quad \mathcal{L}_{\text{SOM}} = \mathbb{E}_{\mathcal{Z}}\Big[\sum_{i=1,i\neq j}^{k}\mathcal{S}_i\big(\phi_{i=j}\big)||\operatorname{sg}[\boldsymbol{z}] - \phi_{i=j}||_2^2\Big], \quad (7)$$

with $\operatorname{sg}[\cdot]$ a gradient blocker that impedes gradient updates to the encoder. Note that for $0 < \beta/\alpha < 1$, the proposed neighbourhood *plus*-kernel is a coarse approximation of the Gaussian kernel. The sum of the reconstruction losses $\mathcal{L}_{\text{recon,cont}}$ and $\mathcal{L}_{\text{recon,disc}}$ from the continuous and discrete decoder, respectively, yield the total task loss $\mathcal{L}_{\text{task}}$, which is combined with the topological loss to create the training objective of the SOM-VAE model.

SOM-VAE-prob (Fortuin et al., 2019) and (T)-DPSOM (Manduchi et al., 2021) are extensions of SOM-VAE. SOM-VAE-prob enforces smoothness over time by adding a transition loss (multiplied by $\gamma$) to optimize a first-order Markov model to learn the node transition probabilities, and a smoothness loss (multiplied by $\tau$ in there work, we will use $\zeta$ here) to minimize the quantization error of highly probable transitions. DPSOM is a probabilistic model, based on a variational autoencoder with a non-degenerate approximate posterior (Kingma & Welling, 2013) with soft cluster assignment and a cluster assignment hardening (CAH) loss (Xie et al., 2016). T-DPSOM additionally incorporates a temporal smoothness loss, and an LSTM, which aims to predict the future latent space. This latter functionality is similar to the future-prediction task that is already naturally embedded in the

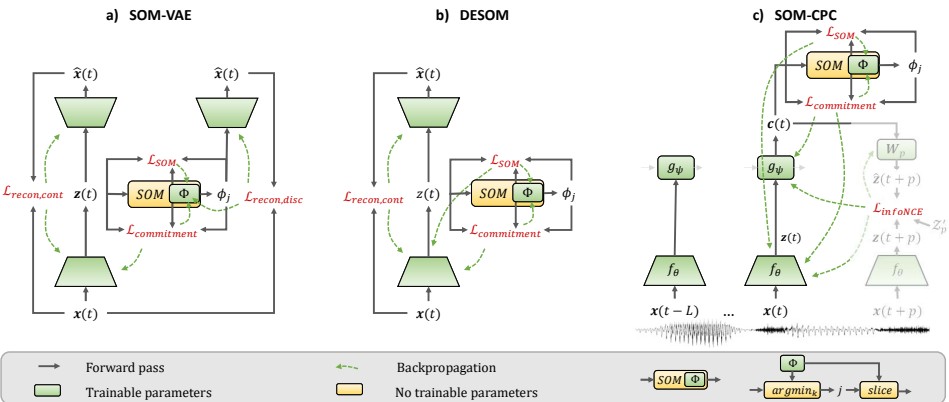

Figure 1: Architectures of different deep-SOM models including the gradient paths in green. a) SOM-VAE (Fortuin et al., 2019) b) DESOM (Forest et al., 2021) c) SOM-CPC (ours). The two decoders in the SOM-VAE model are independent and have their own trainable parameters, while the visualized encoders in the SOM-CPC model are all the same (i.e. parameters are shared). The $g_\psi$ block in the SOM-CPC model indicates an autoregressive component (e.g. a GRU), and $\mathcal{Z}'_p$ refers to a set of drawn negative embeddings.

CPC objective that we propose as a task loss (see section 3.2). The probabilistic additions in the (T-)DPSOM model, with respect to SOM-VAE, are orthogonal to the developments in this work.

Forest et al. (2021) propose Deep embedded SOM (DESOM). Compared to SOM-VAE, the decoder on the discrete space is omitted (therewith also $\mathcal{L}_{\text{recon,disc}}$), gradients from $\mathcal{L}_{\text{SOM}}$ to the encoder are not being blocked (i.e. $\text{sg}[\cdot]$ is removed from eq. (7)), and the topological loss $\mathcal{L}_{\text{topo}}$ is given by eq. (2), i.e. with a Gaussian neighbourhood function with decaying variance. In a short work, Forest et al. (2019) speculate about adding an LSTM in the latent space to train a SOM on sequential data, and refer to this model as LSTM-DESOM. Figure 1a-b visualizes the SOM-VAE and DESOM architecture.

## 3 SOM-CPC

### 3.1 MOTIVATION

In this work, we propose the SOM-CPC model, a representation learning model that learns to map windows of time series data to a structured 2D grid. The model jointly optimizes a temporal contrastive learning objective to extract features, and a topological loss that organizes the SOM space.

In order to learn features that are both suitable for SOM organization *and* accurately reflect the data, the model should ideally invert the original data generating process (which is in general unknown and implicit). Assuming that this generative process has been highly non-linear, feature learning can be formulated as a non-linear independent component analysis (ICA) problem, which has proven to be non-identifiable (Hyvärinen & Pajunen, 1999). However, recent advances showed that the problem becomes identifiable under the assumed presence of an auxiliary variable (Hyvärinen et al., 2018). Such an auxiliary variable (e.g. a temporal component) is not present in plain autoencoders, but the contrastive learning paradigm was shown to conform to this assumption (Hyvärinen et al., 2018; Zimmermann et al., 2021). This theory is in line with the hypothesis stated by Mrabah et al. (2020) that a reconstruction objective may hamper clustering performance in the latent space.

### 3.2 ALGORITHMIC DETAILS

We introduce $\mathcal{X} = \{\ldots, \boldsymbol{x}(t), \boldsymbol{x}(t+1), \ldots\}$, a set of non-overlapping data windows $\boldsymbol{x}(t) \in \mathbb{R}^{ch \times T}$, with $ch$ the number of channels, and $T$ the number of samples in the window. For brevity we omit the time index when possible. An encoder, parameterized by $\theta$, maps each data window $\boldsymbol{x}$ to a latent representation $\boldsymbol{z} = f_\theta(\boldsymbol{x}) \in \mathbb{R}^F$, with $F$ the number of features. The set $\mathcal{Z}$ includes the embeddings of all windows in $\mathcal{X}$. A causal auto-regressive (AR) module $g_\psi$ parameterized by $\psi$, e.g. a gated-recurrent unit (GRU), subsequently aggregates the current and $L$ previous embeddings, to generate a (current) context vector $\boldsymbol{c}(t) \in \mathbb{R}^F$. Given this context, the pretext task in our SOM-CPC

model aims to minimize the prediction error for $P$ future (or 'positive') embeddings $\boldsymbol{z}(t+p)$, for $p \in \{1, \ldots, P\}$, compared to this error for $N$ 'negative' embeddings. These negatives may be sampled across the dataset, or within the same time-series, and are on the fly encoded to their latent representation during training. The task objective, being the InfoNCE loss (Oord et al., 2019), is defined as:

$$\mathcal{L}_{\text{task}} := \mathcal{L}_{\text{InfoNCE}} = \frac{1}{P} \sum_{p=1}^{P} \mathcal{L}_p, \quad \text{with} \quad \mathcal{L}_p = -\underset{\mathcal{X}}{\mathbb{E}} \left[ \log \frac{\exp\left(\boldsymbol{z}(t+p)\mathbf{W}_p \boldsymbol{c}(t)\right)}{\sum_{\boldsymbol{z}' \in \mathcal{Z}'_p \cup \{\boldsymbol{z}(t+p)\}} \exp\left(\boldsymbol{z}'\mathbf{W}_p \boldsymbol{c}(t)\right)} \right], \quad (8)$$

with $\mathcal{Z}'_p \subset \mathcal{Z}$ a set of embeddings of drawn negative samples ($|\mathcal{Z}'_p| = N$), and $\mathbf{W}_p \in \mathbb{R}^{F \times F}$ a trainable mapping between the current context vector and the $p^{\text{th}}$ future embedding.

The context vector is not only used to predict future embeddings, it is also the input to the SOM module that selects the winning node. The SOM is optimized using the topological loss $\mathcal{L}_{\text{topo}}$, as defined in eq. (5), with a Gaussian neighbourhood kernel $\mathcal{S}$, as defined in eq. (2). Depending on the use case, it was found to not always be necessary, or even beneficial (due to higher risk of overfitting), to use an AR module $g_\psi$ to aggregate causal context into the current embedding. If the AR module is not used, the future predictions are made directly from the current (continuous) latent space $\boldsymbol{z}(t)$ instead of $\boldsymbol{c}(t)$. Likewise, $\boldsymbol{z}(t)$ rather than $\boldsymbol{c}(t)$ is being quantized by the SOM module. Depending on the presence of this AR module, both $\mathcal{L}_{\text{topo}}$ and $\mathcal{L}_{\text{task}}$ are thus computed on either $\boldsymbol{z}(t)$ or $\boldsymbol{c}(t)$.

All model elements are optimized jointly, with the training objective being: $\mathcal{L}_{\text{SOM-CPC}} = \mathcal{L}_{\text{task}} + \alpha\mathcal{L}_{\text{topo}}$, which adheres to the general objective of a deep-SOM model as formulated in eq. (4). Figure 1c provides an overview of the SOM-CPC model, and its gradient paths in green. The initial standard deviation $\sigma^{(0)}$ of the Gaussian kernel (from eq. (2)) was set to half the squared-root of the number of SOM nodes $k$. Given the square topology of the SOM grid, this setting of $\sigma_0$ ensures that the full grid is captured by the neighbourhood kernel at the start of training. Algorithm 1 in appendix A.1 provides pseudocode of the full SOM-CPC algorithm.

### 3.3 PERFORMANCE EVALUATION

Forest et al. (2020) provide a taxology of SOM metrics that distinguishes *external* vs *internal* and *topological* vs *clustering* metrics. External metrics are related to labels (which are not used during unsupervised training), while internal metrics do not depend on such information. Topological metrics assess the topological ordering (i.e. neighbourhood relations) of the SOM, while clustering metrics are more related to, for example, pureness of nodes.

To evaluate clustering performance, linked to external labels, we leverage *purity* and the *normalized mutual information* (NMI). The latter corrects for a high number of clusters (i.e. nodes), which could easily lead to high pureness, but leaves the NMI more conservative.

Even though scoring high on external metrics is not the main goal of a representation learning model like SOM-CPC, we do report it as it provides an indication of how well information was preserved. To compute regression/classification performance, we first 'color' (or label) each node with the most occurring (for discrete labels) or median (for continuous labels) label from the training set. The test set predictions are then converted from node indices to label predictions by using these colorings. Regression performance is expressed as the average squared regression error with the target: $\text{SE}_{\text{target}}$. Classification performance is reported with *Cohen's kappa* (Cohen, 1960), a commonly used metric that corrects for correctness by chance.

Topographic performance is measured using the (internal) *topographic error* (TE) (Kiviluoto, 1996), which reports the fraction of windows (between 0 and 1) for which the winning and second-best winning node are not neighbours in the SOM (lower is better). Finally, to measure whether a time series conveys a smooth trajectory through SOM space, we measure the average Euclidean distance (denoted $\ell_{2,\text{smooth}}$) between all subsequent windows in each time series. The lower this value, the less frequently large jumps in the 2D map occur. Note that in extreme cases where many windows collapsed to the same node, both the TE and the average $\ell_{2,\text{smooth}}$ metric are artificially pushed down. We can thus only interpret these metrics in conjunction with earlier-mentioned clustering and classification metrics.

Table 1: Mean and one std. dev. across all test set series of synthetic data. Results for varying values of $\alpha$, $\gamma$, and $\zeta$ can be found in table 3 (appendix A.3.2). SOM-CPC clearly outperforms all baselines. Models below the dashed line are ablations. Regression plots and SOMs of models with a * are visualized in fig. 2a-c.

| | Model | $\alpha$ | $\mathcal{S}$ | $\mathcal{L}_{\text{SOM}}$ sg[·] | $\text{SE}_{\text{target}}$ | $\ell_{2,\text{smooth}}$ | TE |
|---|---|---|---|---|---|---|---|
| | CPC + linear classifier | - | - | - | 2.62±2.37 | - | - |
| | CPC + K-means | - | - | - | 1.09±.62 | - | - |
| | CPC ($F = 2$) + linear classifier | - | - | - | 25.01±42.94 | - | - |
| | CPC ($F = 2$) + K-means | - | - | - | .76±1.31 | - | - |
| | CPC + PCA + linear classifier | - | - | - | 42.81±58.12 | - | - |
| | CPC + PCA + K-means | - | - | - | 4.42±9.01 | - | - |
| * | SOM-VAE | .1 | Plus | ✓ | 8.02±4.58 | 2.41±.68 | .28±.070 |
| | SOM-VAE | 1e-3 | Gaussian | ✓ | 11.60±25.43 | 1.92±.34 | .056±.023 |
| | SOM-VAE-prob | .1 | Plus | ✓ | 20.10±48.80 | 3.15±.73 | .63±.050 |
| * | DESOM | .1 | Gaussian | ✗ | 10.77±10.85 | 2.20±.44 | .061±.019 |
| * | SOM-CPC (ours) | 1e-4 | Gaussian | ✗ | .72±1.08 | 1.37±.37 | **.022±.011** |
| Ablations | SOM-CPC | 1e-2 | Gaussian | ✓ | **.47±.48** | **.99±.24** | .069±.040 |
| Ablations | SOM-CPC | 1e-4 | Plus | ✗ | 1.71±1.15 | 2.46±0.51 | .30±0.062 |
| Ablations | SOM-CPC | 1e-2 | Plus | ✓ | 1.16±.61 | 1.85±.26 | .12±.037 |
| Ablations | CPC + SOM (disjoint) | - | Gaussian | - | .84±1.14 | 1.47±.50 | .028±.014 |

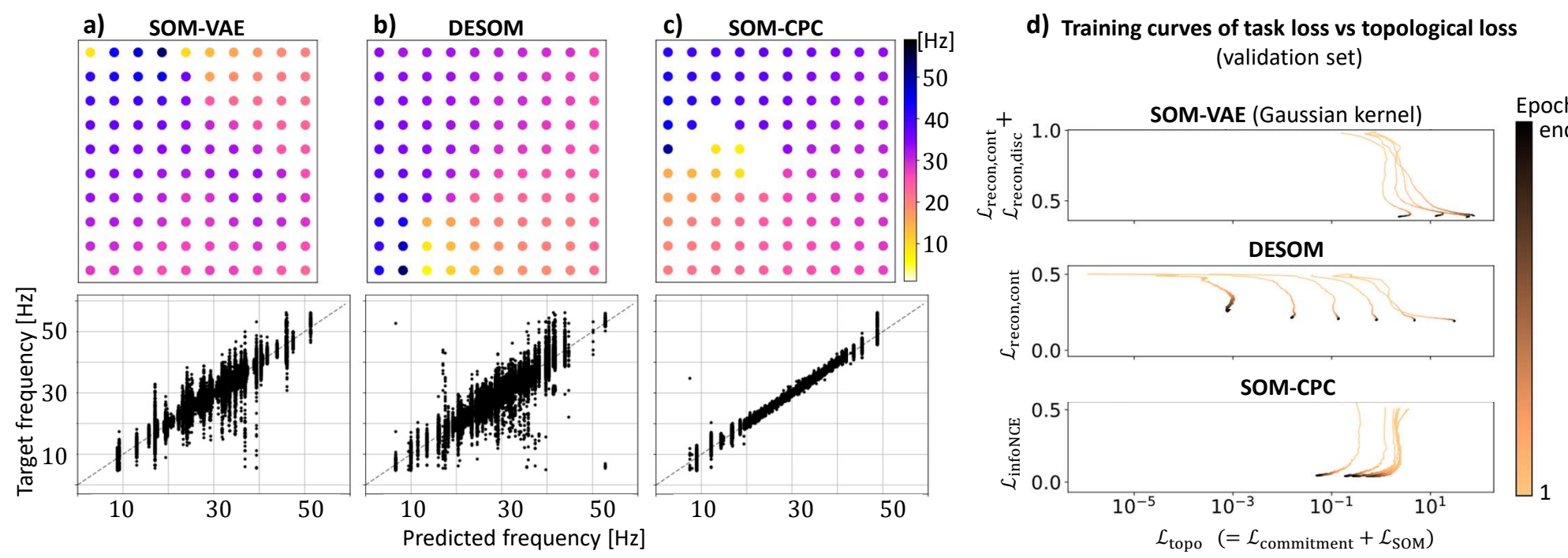

Figure 2: SOMs and regression plots for SOM-VAE (a), DESOM (b) and SOM-CPC (c). Both DESOM and SOM-CPC show a gradual change of frequency over the grid, but the regression error $\text{SE}_{\text{target}}$ is lower for SOM-CPC, which can also be seen from the regression plot, where the predicted window frequencies are plotted against the target frequencies (i.e. training set median label) for the node on which the window was mapped. d) Task loss versus the topological loss for SOM-VAE (with Gaussian neighbourhood), DESOM and SOM-CPC (both with and without sg[·]). The different curves display various values of $\alpha$, for which the DESOM model seems most sensitive. The SOM-CPC models follow a smooth optimization curve, minimizing both the task and topological loss, while these losses seem to be more conflicting in SOM-VAE and DESOM training.

# 4 EXPERIMENTS

We compare SOM-CPC to several other 2D representation learning methods. First, deep-SOM models with a reconstruction task loss (i.e. SOM-VAE, SOM-VAE-prob, and (GRU-)DESOM). Second, vanilla CPC with a multi-dimensional latent space ($F \gg 2$) (Oord et al., 2019), followed by PCA for additional dimensionality reduction to 2D. Third, CPC with a 2D latent space ($F = 2$). For the latter two CPC-based models, linear and non-linear read-out is, respectively, tested using a linear neural classifier, and K-means clustering with the same number of clusters as the number of nodes used in SOM-CPC. High-dimensional vanilla CPC ($F \gg 2$) without additional dimensionality reduction is, moreover, tested as well as it sets a baseline for the amount of information that can be preserved given the encoder architecture, while not providing an interpretable 2D representation. The same encoder architecture is used for all models that are compared in a single application domain, and all models are run with the same seed for randomization. Details on model architectures and training settings for the different applications can be found in appendix A.3.1, A.4.2, and A.5.1.

## 4.1 SYNTHETIC DATA

**Data generation:** A synthetic dataset was created, consisting of sinusoids with an initial frequency sampled from a uniform distribution between 20 and 40 Hz. The frequency of the signals was altered over time according to a random walk process with a step size of 0.1 Hz. As such, at each time step (i.e. sample), the signal's frequency either increased with 0.1 Hz (with probability $p_{\text{up}} = 0.1$),

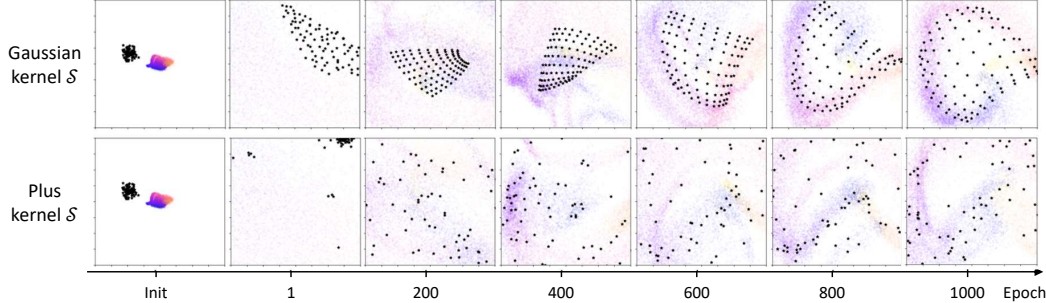

Figure 3: Progression of SOM training (nodes are indicated in black) in the SOM-CPC model, using either a Gaussian (top) or plus neighbourhood (bottom) kernel. The PCA projection of the test set latent space is plotted behind the nodes. It can clearly be seen that the Gaussian kernel enforces a more strict organization of SOM nodes, where nodes are non-uniformly quantizing the latent space, placing more nodes at higher density areas.

decreased with 0.1 Hz ($p_{\text{down}} = 0.1$), or remained constant ($p_c = 0.8$). In case the random walk crossed either 1 or 60 Hz, the probabilities were (temporarily) altered to $[p_{\text{up}}, p_c, p_{\text{down}}] = [0.5, 0.5, 0]$ or $[p_{\text{up}}, p_c, p_{\text{down}}] = [0.0, 0.5, 0.5]$, respectively. All series were finally corrupted with an additive white Gaussian noise vector $\epsilon \sim \mathcal{N}(0, 0.01)$. Formally, each generated signal took the form: $\boldsymbol{x}[n] = \sin\left(2\pi \frac{f[n-1]+\Delta f}{f_s} n\right) + \epsilon$, with $f[n=0] \sim U[20, 40]$, $\Delta f \sim \text{Categorical}([p_{\text{up}}, p_c, p_{\text{down}}])$, and $f_s = 128$ Hz the sampling frequency. A total of 200 of such time-series, each of 5 minutes, were generated, and labels were defined per 1-second window by taking the median frequency. The set was randomly divided into a training ($n = 100$), validation ($n = 50$), and test split ($n = 50$).

**Results:** Table 1 shows that SOM-CPC outperforms all deep-SOM baselines on all metrics. Figure 6 in appendix A.3.2 shows the PCA projections of the (continuous) latent spaces of the three deep-SOM models indicated with a $*$ in table 1. It reveals that the latent space disentanglement of the SOM-CPC model is much better than that of the SOM-VAE and DESOM models. Figure 2a-c displays the resulting SOMs (colored with the median test set labels) for the same three models. Uncolored nodes in the SOM were not assigned in the test set. Interestingly, although the SOM for the DESOM and SOM-CPC model look similar, the $\text{SE}_{\text{target}}$ is higher for the DESOM model, which can also be seen from the regression plots below the SOMs.

The addition of two temporal losses in the SOM-VAE-prob model, as compared to SOM-VAE, did deteriorate the given metrics, even though a range of values for multipliers $\alpha$, $\gamma$ and $\zeta$ was tested (see table 3 in appendix A.3.2 for the full sweep). The deterioration of the results can be explained by the difficulty of finding the correct scaling factors for these additional losses. Note that the SOM-CPC model automatically incorporates smoothness over time thanks to the nature of the CPC task loss, therewith preventing additional hyperparameter tuning.

Additionally, we study the optimization behavior of SOM-VAE, DESOM, and SOM-CPC by plotting the progression of the task versus the topological loss during training (see fig. 2d). To make a fair comparison, we plot the SOM-VAE models that are trained with a Gaussian neighbourhood kernel. Different curves in the graphs indicate runs with varying values for $\alpha$, and the line color's gradient denotes the training iteration. The SOM-CPC graphs include the models run with and without gradient detachment of $\mathcal{L}_{\text{SOM}}$ to the encoder. It can be seen that both losses jointly minimize in SOM-CPC training, while there is a counteracting effect visible for SOM-VAE, and a high influence of the value of $\alpha$ for DESOM training.

Comparing to non-deep-SOM baselines, it can be seen from table 1 that CPC (with $F = 2$), and CPC followed by PCA, resulted in a much higher regression error $\text{SE}_{\text{target}}$ than SOM-CPC when using linear read-out. Non-linear K-means clustering improved performance for both cases, but only for CPC with $F = 2$, performance nearly reached SOM-CPC performance. Later we will see that optimizing CPC with $F = 2$ can hamper optimization for more intricate data spaces (see section 4.3). Interestingly, regression performance of SOM-CPC was found to be even slightly better than that of the vanilla multi-dimensional CPC model (with $F = 128$), both for linear classification and K-means. This could be explained by the additional regularization that the SOM provides in SOM-CPC training.

We perform several ablation experiments on SOM-CPC, which are reported below the dashed line in table 1. Blocking the gradients of the neighbour nodes with respect to the encoder during training ($\mathcal{L}_{\text{SOM}}$ sg[·] column) slightly improved the regression error and temporal smoothness, but decreased the topographic error. Looking at the models with various values for $\alpha$ (reported in table 3, appendix A.3.2), the effect of gradient blocking can be considered small and ambiguous when considering different metrics. Disjoint training (CPC + SOM) resulted in a less smooth trajectory over time through the 2D SOM space, seen from the higher $\ell_{2,\text{smooth}}$. Using a plus neighbourhood kernel instead of a Gaussian kernel decreased performance on all three metrics. The increase in TE, is well explainable by the fact that a plus kernel takes into account fewer neighbours (at least at the start of training) and therefore has more difficulty to find a good topological mapping. Figure 3 shows the development of the SOM node spread (projected on top of a PCA projection of the continuous test set latents), during training for SOM-CPC with the two type of kernels. It can indeed be seen that the Gaussian kernel enforces a more strict topological organization. Interestingly, the kernel does not only influence the codebook vectors, but also seems to influence the organization of the latent space, seen from the differently-shaped PCA projections in the background.

## 4.2 SLEEP

We analyse SOM-CPC on subset 3 of the Montreal Archive of Sleep Studies (MASS) database (O'Reilly et al., 2014), consisting of whole-night polysomnography recordings, for which every 30-second window is labelled with a sleep stage label from {N1, N2, N3, REM, Wake}. The 62 recordings (from 62 unique subjects) were randomly split into a training ($n = 48$), validation ($n = 8$) and hold-out test set ($n = 7$). Details on the data preprocessing can be found in appendix A.4.1.

Table 5 in appendix A.4.3 shows that SOM-CPC again clearly outperformed SOM-VAE and DESOM on all metrics. Whether or not the gradients of the SOM loss were stopped towards the encoder did not greatly influence SOM-CPC performance. Topological ordering, measured by TE, and temporal smoothness ($\ell_{2,\text{smooth}}$) deteriorated when changing the Gaussian kernel to a plus kernel, or training high-dimensional CPC and SOM disjointly. SOM-CPC's classification performance was higher than that of CPC with $F = 2$, and CPC followed by PCA.

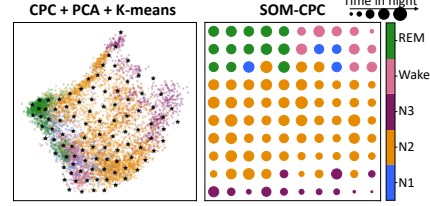

Figure 4: Deep sleep N3 is isolated from light sleep N1, Wake and REM sleep with a cluster of medium-deep sleep N2.

Figure 4 shows the test set PCA projection of the latent space of CPC ($F = 128$), with the K-means nodes as black stars (left), and the SOM (right) trained by the SOM-CPC model (nodes are colored with the most-occurring label in the test set). Both visualizations show similar clustering patterns: deep sleep N3 is isolated from lighter forms of sleep (i.e. N1, Wake and REM sleep) by a thick cluster of medium-deep sleep N2. However, the higher performance of SOM-CPC (see table 5) indicates that more information is preserved in the 2D space resulting from the SOM-CPC model. The size of the nodes in the SOM map of SOM-CPC indicates the average time in the night of windows on that node. A difference is visible in node sizes within the Wake, N2 and N3 clusters, suggesting a possible existence of different sub-categories of sleep within the pre-defined sleep stages.

## 4.3 AUDIO

For the audio experiments, we use a subset of the publicly available LibriSpeech dataset (Panayotov et al., 2015). The dataset contains multiple minute-long English voice recordings of 251 different speakers, sampled at 16 KHz. We used the publicly available train-test split, as provided by Oord et al. (2019), and created an additional validation set by randomly selecting 25% of the training set. Recordings of the ten speakers with the longest recording time were selected to alleviate computational burden. This resulted in a total of 150.9, 54.6, and 46.5 minutes in the training, validation, respectively test set. The full model and training details can be found in appendix A.5.1.

Table 7 in appendix A.5.2 shows the results of SOM-CPC (which includes a GRU for this dataset), compared to different variants of the DESOM model. SOM-CPC outperforms all DESOM variants by a wide margin and for all choices of the $\alpha$ parameter. The difference in performance between DESOM and SOM-CPC is also visible in fig. 5. SOM-CPC has clustered the SOM nodes belonging

to the same speaker, and seems to group male and female speakers (denoted with the node's shape), while these effects are not present in the SOM of the GRU-DESOM model. Minimizing the InfoNCE training objective of CPC with $F = 2$ was found challenging for this dataset, which resulted in non-competitive performance of the linear classifier and K-means clustering trained on the resulted 2D latent space. Using PCA for dimensionality reduction of the high-dimemnsional CPC latent space (with $F = 512$) performed better, but still inferior to SOM-CPC.

The SOM of the SOM-CPC model (fig. 5-right) reveals two separate clusters both for speaker 2 (green) and 3 (red). The LibriSpeech corpus contains multiple recordings from each speaker, grouped by (book) chapters from which the speaker was reading. Interestingly, additional analyses revealed that the red and green sub-clusters represented recordings belonging to different chapters: $99.99\%$ of the test-set windows mapped to the upper red sub-cluster belong to the same chapter, while $99.97\%$ of the windows in the lower red sub-cluster belong to another chapter read by this speaker. Similarly, $100.0\%$ of the test set windows

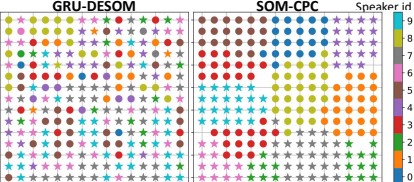

Figure 5: SOM-CPC is able to better cluster different speakers than GRU-DESOM. Stars denote women and dots are men.

in the right green sub-cluster belong to two chapters read by speaker 2, while $98.91\%$ of the windows in the left green sub-cluster belong to another chapter. An auditory inspection revealed that the room acoustics of the recordings belonging to the chapters in different clusters were different, causing changes in the signals which the SOM-CPC model has picked upon. This division between recordings of the same speaker is not visible in the 2D PCA projection of the CPC (with $F = 128$) features, as seen from fig. 8 in appendix A.5.2.

## 5  DISCUSSION

We proposed a new member of the deep-SOM family: SOM-CPC, suitable for interpretable 2D representation learning of high-rate data streams. Earlier proposed deep-SOM models mainly used reconstruction objectives. In general, SOM-CPC outperformed these models with a wide gap on a variety of metrics. Moreover, it implicitly enforces temporal smoothness, while autoencoder-based models require additional losses and hyperparameter tuning to achieve this. SOM-CPC's task loss was found to align better with the topological SOM objective than a reconstruction loss, as already hypothesized by Mrabah et al. (2020). While for some applications CPC could succesfully be trained with a 2D latent space directly, optimization was found to be hampered in case of more intricate data spaces. Compared to vanilla CPC with a multi-dimensional latent space, SOM-CPC enables pattern recognition and knowledge discovery. The SOM objective did not hamper CPC optimization. Even better, in the synthetic setup it had a regularizing effect, resulting in lower regression error than vanilla CPC. The use of a Gaussian neighbourhood kernel, as opposed to a plus kernel, was found to improve the topological ordering in the SOM. No decisive conclusions could be made regarding gradient blocking from the SOM loss towards the encoder parameters. Allowing these gradients to flow did not hurt performance, so for coding simplicity, we would advice to not detach the SOM loss.

Setting an appropriate stopping criterion for self-supervised (SSL) models is debatable. In the SSL literature models with the best test set performance are sometimes reported (He et al., 2019; Fortuin et al., 2019). This is, however, questionable as it may artificially boost reported performance. As such, we created a validation set to apply early stopping in all experiments. Another challenge arises when dealing with aggregated loss functions, since not all losses may smoothly decay and the weighted summation of losses may result in a different optimal epoch than the sub-losses separately. Besides, classification performance (often used as a proxy for information preservation) does not necessarily align with SOM performance or information preservation (see fig. 7 in appendix A.4.3).

We believe that SOM-CPC will facilitate knowledge discovery in real-life time series and opens up new research directions for representation learning of time series. Directions include investigation to whether additions like the soft-cluster assignment, cluster hardening loss or a Gaussian latent prior - which have shown to improve the SOM-VAE model (Manduchi et al., 2021) - improve SOM-CPC performance as well. Moreover, the CPC objective assumes slowly (or non-changing) data characteristics within the time frame in which positive samples are drawn. A multi-modal variational future prediction could possibly improve performance for data that do not meet this assumption.

## REPRODUCIBILITY STATEMENT

All code used to train and evaluate the models as presented in this paper can be found at `https://anonymous.4open.science/r/SOM-CPC`. The details regarding model architectures and training settings for each of the application domains are also presented in appendix A.3.1, A.4.2, and A.5.1. Pseudocode of the proposed SOM-CPC algorithm is given in algorithm 1 in appendix A.1.

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

# A    EXPERIMENTAL DETAILS

This appendix contains all information for full reproducability of the experiments. Domain-independent details on SOM-CPC and its evaluation are provided in appendix A.1, while benchmark implementations are discussed in appendix A.2. Domain-specific settings for the synthetic, sleep and audio experiments are discussed in sections A.3, A.4, and A.5, respectively.

## A.1    GENERAL DETAILS

Algorithm 1 provides pseudocode of SOM-CPC, when considering the presence of an AR module. For reproducibility, the full code base can be found at `https://anonymous.4open.science/r/SOM-CPC`.

---

**Algorithm 1** SOM-CPC

---

**Input:** Dataset $\mathcal{X}$, model comprising $f_\theta$, $g_\psi$, $\{\mathbf{W}_p\}_{p=1}^P$, number of past windows $L$, # positive samples $P$,
   # negative samples $N$, # SOM nodes $k$, Gaussian neighbourhood function $\mathcal{S}$ with $\sigma^{(n_{\max})}$, $n_{\max}$, loss trade-off
   parameter $\alpha$.
**Output:** A trained SOM: topologically ordered codebook $\Phi$ that represents data $\mathcal{X}$ in 2D.
   - Initialize the Gaussian kernel according to eq. (2) with: $\sigma^{(0)} = \frac{1}{2}\sqrt{k}$  and  $\lambda = -n_{\max}/\log(\sigma^{(n_{\max})}/\sigma^{(0)})$
  **for** $n$ in $n_{\max}$ **do**
     - Sample a sequence of datapoints: $[\boldsymbol{x}(t-L), \ldots, x(t)] \sim \mathcal{X}$
     - Define $P$ positive samples: $\{\boldsymbol{x}(t+p)\}_{p=1}^P$
     - Sample $P \times N$ negative samples $\mathcal{X}' \subset \mathcal{X}$, with $|\mathcal{X}'| = P \times N$
     - Encode
       - data sequence: $\boldsymbol{c}(t) = g_\psi\Big(f_\theta\big([\boldsymbol{x}(t-L), \ldots, x(t)]\big)\Big)$
       - positive samples: $\{\boldsymbol{z}(t+p)\}_{p=1}^P = f_\theta\big(\{\boldsymbol{x}(t+p)\}_{p=1}^P\big)$
       - negative samples: $\mathcal{Z}' = f_\theta(\mathcal{X}')$
     - Predict future: $\hat{\mathbf{z}}(t+p) = \mathbf{W}_p\boldsymbol{c}(t)$, with $1 \leq p \leq P$
     - Quantize: $\phi_{i=j} = \text{SOM}_\Phi\big(\boldsymbol{c}(t)\big)$
     - Update $\sigma^{(n)}$ according to eq. (3)
     - Compute $\mathcal{L}_{\text{topo}}\big(\boldsymbol{c}(t)\big)$ and $\mathcal{L}_{\text{infoNCE}}$ according to eq. (5) and eq. (8)
     - Update trainable parameters $\propto \alpha\mathcal{L}_{\text{topo}}$ and $\mathcal{L}_{\text{infoNCE}}$
  **end for**

---

Several metrics were used to quantify SOM performance, as explained in section 3.3. The purity implementation is taken from the Github implementation of Fortuin et al. (2019), while NMI was computed using the sklearn library (Pedregosa et al., 2011). The topographic error implementation comes from the *SOMperf* python library (Forest et al., 2020). Finally, the $\ell_{2,\text{smooth}}$ distance is computed using the *norm* function in the *linalg* library of numpy.

Section 3.3 already shortly elucidated upon the way in which Cohen's kappa and $\text{SE}_{\text{target}}$ were computed. For both metrics, each SOM node was labeled/colored with the most-occuring (in case of Cohen's kappa) or median (in case of $\text{SE}_{\text{target}}$) label in the training set. Note that it can be questioned whether this node labelling should be done on a training or validation set, or directly on the test set for which performance is reported. In earlier times when more conventional clustering approaches (e.g. K-means) were used, a training/validation/test split was typically not made. As a result, clustering was directly performed on the one and only (test) set, that was also used to label the clusters/nodes. Moving towards deep learning based approaches where more hyperparameters need to be set and overfitting can become a larger problem, we found it necessary to report in this work on a test set that was not used for labelling the nodes and/or setting hyperparameters. It should thus be taken into account, that direct comparison to results in other deep-clustering works may need a critical eye to see whether similar procedures were used or not.

## A.2    BENCHMARKS AND ABLATIONS

To benchmark our implementation of the SOM-VAE model (and the very similar DESOM model), we replicated the results on MNIST. MNIST was not used for further experimentation with SOM-CPC in the main body of this paper since this work focuses on high-rate time series. Using $k = 16$ SOM

nodes, Fortuin et al. (2019) report purity $= 0.731 \pm 0.004$ and NMI $= 0.594 \pm 0.004$ in table 1, and purity $= 0.721 \pm 0.006$ and NMI $= 0.587 \pm 0.003$ in table S1. All numbers are averages and standard errors over 10 runs.

Settings that were not provided in the paper, were taken from the hard-coded settings that we found in the provided code base. Instead of splitting the standard MNIST training set in a training and test split, as done by Fortuin et al. (2019), we used the available train/test split that comes with the standard MNIST dataloader from Pytorch. From the first ten runs, one run fully collapsed and resulted in extremely poor performance. Considering this run as an outlier, we run an $11^{\text{th}}$ run and here report the average and standard errors of the 10 non-collapsed runs: purity $= 0.705 \pm 0.002$ and NMI $= 0.584 \pm 0.001$. Our performance does reasonably well match the reported performance by Fortuin et al. (2019), given the fact that a different test set of MNIST was used for our experiments.

To restrict the search space of hyperparameters in the SOM-VAE(-prob) model, which has multiple loss multipliers, we fixed some of the these multipliers for further experiments in this paper. Multiplier $\beta$ scales the SOM loss that sums the quantization error of the four neighbour nodes in the plus kernel. It is, therefore, expected to be at least 4 times larger than the commitment loss, which only reflects the quantization error of the winning node. Choosing $\beta = \alpha/4$ would imply that the weighing of the summed quantization error of the four neighbours is equal to the weighing of this error for the winning node. To give the winning node a slightly higher importance, we set $\beta = \alpha/5 = 0.2\alpha$. The authors of SOM-VAE (Fortuin et al., 2019) used, instead, a search strategy to find the optimal setting. Their code base[1] shows that the SOM loss was multiplied with $0.9$, while $\alpha = 1$. Taking into account that their implementation of the commitment loss averaged the quantization error of the four neighbour nodes, while we summed the contribution, their effective setting was thus set to $\beta = \frac{0.9}{4}\alpha = 0.225\alpha$, which is close to what we used in our experiments.

We compared SOM-CPC also against vanilla CPC training followed by a linear classifier or K-means clustering, while freezing the encoder parameters. The supervised linear classifier took in all experiments the form of one fully-connected layer, including biases, that was followed by a log-softmax activation for the sleep and audio cases. It was trained using the mean squared error for the synthetic data set, and cross-entropy loss for sleep and audio experiments. K-means clustering was run from the sklearn library, with the default settings. The number of clusters was chosen to be equal to the number of nodes in the SOM-CPC models against which the performance was compared. Also the disjointly-trained SOM had exactly the same settings as the SOM in the SOM-CPC models with which it was compared.

Several ablation are performed on the SOM-CPC model. We test the effect of propagating gradients of $\mathcal{L}_{\text{SOM}}$ to the encoder parameters, the difference between using a Gaussian neighbourhoood kernel versus a plus kernel, and the effect of jointly training CPC and the SOM. Moreover, the effect of certain settings that are typically used in the CPC objective are investigated. CPC's InfoNCE objective for one window $p$, given in eq. (8), can as follows be generalized to a more general contrastive learning objective:

$$\mathcal{L}_p = -\mathbb{E}_{\mathcal{X}}\left[ \log \frac{\exp\left( \text{sim}(\boldsymbol{z}_a, \boldsymbol{z}_p)/\tau \right)}{\sum_{\boldsymbol{z}' \in \mathcal{Z}'_p \cup \{\boldsymbol{z}_p\}} \exp\left( \text{sim}(\boldsymbol{z}_a, \boldsymbol{z}'_p)/\tau \right)} \right], \tag{9}$$

where $\boldsymbol{z}_a$ is the latent space of the current (or *anchor*) window, $\text{sim}(\cdot)$ a similarity metric, and the other symbols are equivalent to eq. (8). CPC typically uses $\tau = 1$, and the dot product as the similarity metric. However, other related contrastive learning objectives, e.g. in SimCLR (Chen et al., 2020), use a temperature value that is often set to $0.07$ (Chen et al., 2020; Woo et al., 2022), and a cosine similarity instead of the (unnormalized) dot product. As such, we add ablations where we set $\tau$ at $1$ or $0.07$, and use either the dot-product or the cosine similarity as the similarity metric.

## A.3 SYNTHETIC EXPERIMENTS

### A.3.1 TRAINING DETAILS

Table 2 summarizes the encoder and decoder architectures used in this experiment. The *output size* column in the table uses channels-first notation. The SOM-VAE and DESOM models were

---

[1] `https://github.com/ratschlab/SOM-VAE/blob/master/som_vae/somvae_train.py`, line 79.

Table 2: Model details for the synthetic data experiments in section 4.1.

| Layer type | Output size | Channels | Activation | Kernel size | Strides | Dilation | Padding |
|---|---|---|---|---|---|---|---|
| **Encoder for SOM-CPC and CPC** | | | | | | | |
| Conv1D | bs $\times 16 \times 128$ | 16 | Leaky ReLU (0.01) | 9 | 1 | 1 | same |
| MaxPool1D | bs $\times 16 \times 32$ | - | - | 4 | 4 | - | - |
| Dropout (0.1) | bs $\times 16 \times 32$ | - | - | - | - | - | - |
| Conv1D | bs $\times 32 \times 32$ | 32 | Leaky ReLU (0.01) | 7 | 1 | 1 | same |
| MaxPool1D | bs $\times 32 \times 8$ | - | - | 4 | 4 | - | - |
| Dropout (0.1) | bs $\times 32 \times 8$ | - | - | - | - | - | - |
| Conv1D | bs $\times 64 \times 8$ | 64 | Leaky ReLU (0.01) | 3 | 1 | 1 | same |
| MaxPool1D | bs $\times 64 \times 2$ | - | - | 4 | 4 | - | - |
| Dropout (0.1) | bs $\times 64 \times 2$ | - | - | - | - | - | - |
| Conv1D | bs $\times 128 \times 2$ | 128 | Leaky ReLU (0.01) | 3 | 1 | 1 | same |
| MaxPool1D | bs $\times 128 \times 1$ | - | - | 2 | 2 | - | - |
| **Encoder for SOM-VAE and DESOM** | | | | | | | |
| Conv1D | bs $\times 16 \times 128$ | 16 | Leaky ReLU (0.01) | 9 | 1 | 1 | same |
| MaxPool1D | bs $\times 16 \times 32$ | - | - | 4 | 4 | - | - |
| Dropout (0.1) | bs $\times 16 \times 32$ | - | - | - | - | - | - |
| Conv1D | bs $\times 32 \times 32$ | 32 | Leaky ReLU (0.01) | 7 | 1 | 1 | same |
| MaxPool1D | bs $\times 32 \times 8$ | - | - | 4 | 4 | - | - |
| Dropout (0.1) | bs $\times 32 \times 8$ | - | - | - | - | - | - |
| Conv1D | bs $\times 64 \times 8$ | 64 | Leaky ReLU (0.01) | 3 | 1 | 1 | same |
| Flatten | bs $\times 512$ | - | - | - | - | - | - |
| Fully Connected | bs $\times 128$ | 128 | Leaky ReLU (0.01) | - | - | - | - |
| **Decoder for SOM-VAE and DESOM** | | | | | | | |
| Fully Connected | bs $\times 512$ | 512 | Leaky ReLU (0.01) | - | - | - | - |
| Unflatten | bs $\times 64 \times 8$ | - | - | - | - | - | - |
| Conv1D | bs $\times 32 \times 8$ | 32 | Leaky ReLU (0.01) | 3 | 1 | 1 | same |
| ConvTranspose1D | bs $\times 32 \times 32$ | 32 | None | 4 | 4 | 1 | 0 |
| Conv1D | bs $\times 16 \times 32$ | 16 | Leaky ReLU (0.01) | 7 | 1 | 1 | same |
| ConvTranspose1D | bs $\times 16 \times 128$ | 16 | None | 4 | 4 | 1 | 0 |
| Conv1D | bs $\times 1 \times 128$ | 1 | Tanh | 9 | 1 | 1 | same |

found to benefit from a convolutional part of the encoder that did not fully reduce the temporal dimension to size 1. As such, the last convolutional layer of the SOM-CPC encoder was changed for a fully connected layer preceded by a flattening operation for the autoencoder-based models. The SOM-CPC and CPC model are run without an AR module, to make the fairest comparison to the SOM-VAE(prob) and DESOM models, which also do not incorporate such a component.

For SOM-CPC, $P = 3$ future predictions (i.e. positive samples) were used, and $N = 3$ negative samples were drawn for each positive sample. The latter were drawn randomly from the entire training set. The standard deviation of the Gaussian neighbourhood kernel was exponentially decayed until $\sigma^{(n_{\max})} = 2$. Choosing a lower value at the end of training induced instable optimization behavior.

All models (including the benchmarks) were trained using the Adam optimizer (Kingma & Ba, 2014), with a learning rate of 0.001 and a batch size of 128. Each model was trained for maximally 1000 epochs. The best model was selected based on the lowest task loss on the validation set, being $\mathcal{L}_{\text{recon}}$ for SOM-VAE and DESOM and $\mathcal{L}_{\text{InfoNCE}}$ for SOM-CPC and CPC. We did not use the full training objective $\mathcal{L}_{\text{deep-SOM}}$ as model selection criterion, as both the commitment and SOM loss showed to be low initially (possibly due to low values of the random initialization of the model), while both increased and reached a steady-state later in training. The linear classifier and the disjointly-trained SOM on the CPC embeddings were trained until convergence, for maximally 1000 epochs.

### A.3.2 EXTENDED RESULTS

Table 3 extends table 1 with additional sweeps of hyperparameters $\alpha$, $\gamma$, and $\zeta$, and ablations with different settings of the temperature value $\tau$ and the used similarity metric. Compared to SOM-VAE, SOM-VAE-prob is expected to show more smooth trajectories over time, captured in the $\ell_{2,\text{smooth}}$ distance metric, thanks to the additional transition and smoothness loss (multiplied by $\gamma$ and $\zeta$, respectively). It can be seen that tuning these hyperparameters is a complex process, and a sweep did not result in one SOM-VAE-prob run that performed better than the best SOM-VAE model. In contrary, the addition of the extra losses possibly interfered with the optimization process, and only very delicate settings of $\gamma$ and $\zeta$ might improve model performance eventually. A sweep over the topological loss multiplier $\alpha$, revealed a low sensitivity of SOM-CPC to this value. It can be seen that changing the temperature value and/or the similarity metric did not significantly alter the performance consistently on all reported metrics.

Table 3: This is the extended version of table 1 on the synthetic data results, including a sweep of hyperparameters $\alpha$, $\gamma$ and $\zeta$. Bold values indicate the best performance per column (excluding the upper bound of the vanilla CPC model, which does not result in a 2D representation). The models indicated with a $*$ were used to depict trained SOMs in fig. 2 and PCA projections in fig. 6.

| | Model | | $\alpha$ | $\mathcal{S}$ | $\mathcal{L}_{\text{SOM}}$ sg[·] | $\text{SE}_{\text{target}}$ | $\ell_{2,\text{smooth}}$ | TE |
|---|---|---|---|---|---|---|---|---|
| | CPC + linear classifier | | - | - | - | 2.62±2.37 | - | - |
| | CPC + K-means | | - | - | - | 1.09±.62 | - | - |
| | CPC ($F = 2$) + linear classifier | | - | - | - | 25.01±42.94 | - | - |
| | CPC ($F = 2$) + K-means | | - | - | - | .76±1.31 | - | - |
| | CPC + PCA + linear classifier | | - | - | - | 42.81±58.12 | - | - |
| | CPC + PCA + K-means | | - | - | - | 4.42±9.01 | - | - |
| | SOM-VAE | | 1e-3 | Plus | ✓ | 11.59±13.69 | 2.60±.46 | .38±.042 |
| | | | 1e-2 | Plus | ✓ | 14.57±44.10 | 2.98±.84 | .38±.052 |
| * | | | .1 | Plus | ✓ | 8.02±4.58 | 2.41±.68 | .28±.070 |
| | | | 1 | Plus | ✓ | 13.43±3.66 | 2.75±.45 | .33±.048 |
| | SOM-VAE | | 1e-5 | Gaussian | ✓ | 11.12±17.05 | 1.93±.29 | .069±.029 |
| | | | 1e-4 | Gaussian | ✓ | 1.52±26.61 | 1.95±.36 | .075±.028 |
| | | | 1e-3 | Gaussian | ✓ | 11.60±25.43 | 1.92±.34 | .056±.023 |
| | | | 1e-2 | Gaussian | ✓ | 18.13±48.66 | 2.03±.42 | .086±.030 |
| | SOM-VAE-prob | ($\gamma$ = 5e-5, $\zeta$ = 1e-3) | 1e-3 | Plus | ✓ | 21.26±55.00 | 3.78±.52 | .93±.042 |
| | | ($\gamma$ = 4e-5, $\zeta$ = 1e-3) | 1e-3 | Plus | ✓ | 21.30±37.37 | 4.23±.57 | .88±.051 |
| | | ($\gamma$ = 3.3e-5, $\zeta$ = 1e-3) | 1e-3 | Plus | ✓ | 22.52±48.97 | 3.81±.51 | .98±.015 |
| | | ($\gamma$ = 5e-4, $\zeta$ = 1e-2) | 1e-2 | Plus | ✓ | 14.65±19.58 | 3.70±.51 | .86±.083 |
| | | ($\gamma$ = 4e-4, $\zeta$ = 1e-2) | 1e-2 | Plus | ✓ | 27.25±72.82 | 3.54±.67 | .94±.022 |
| | | ($\gamma$ = 3.3e-4, $\zeta$ = 1e-2) | 1e-2 | Plus | ✓ | 21.62±38.78 | 3.71±.76 | .97±.014 |
| | | ($\gamma$ = 5e-5, $\zeta$ = 1e-2) | .1 | Plus | ✓ | 26.82±68.84 | 3.23±.80 | .68±.067 |
| | | ($\gamma$ = 4e-5, $\zeta$ = 1e-2) | .1 | Plus | ✓ | 22.34±64.92 | 3.11±.73 | .74±.072 |
| | | ($\gamma$ = 3.3e-5, $\zeta$ = 1e-2) | .1 | Plus | ✓ | 2.10±48.80 | 3.15±.73 | .63±.050 |
| | | ($\gamma$ = 5e-4, $\zeta$ = .1) | .1 | Plus | ✓ | 4.85±75.96 | 3.06±.55 | .84±.050 |
| | | ($\gamma$ = 4e-4, $\zeta$ = .1) | .1 | Plus | ✓ | 29.96±46.96 | 2.81±.34 | .82±.13 |
| | | ($\gamma$ = 3.3e-4, $\zeta$ = .1) | .1 | Plus | ✓ | 3.96±56.94 | 3.95±.83 | .85±.11 |
| | DESOM | | 1e-5 | Gaussian | ✗ | 14.00±3.10 | 1.99±.36 | .13±.069 |
| | | | 1e-4 | Gaussian | ✗ | 19.09±44.04 | 1.95±.28 | .11±.046 |
| | | | 1e-3 | Gaussian | ✗ | 12.58±27.10 | 1.92±.31 | .077±.033 |
| | | | 1e-2 | Gaussian | ✗ | 13.66±44.71 | 1.89±.33 | .065±.028 |
| * | | | .1 | Gaussian | ✗ | 10.77±10.85 | 2.20±.44 | .061±.019 |
| | | | 1 | Gaussian | ✗ | 22.86±55.71 | 2.26±.43 | .092±.034 |
| | SOM-CPC (ours) | | 1e-5 | Gaussian | ✗ | .95±2.40 | 1.24±.31 | .048±.020 |
| * | | | 1e-4 | Gaussian | ✗ | .72±1.08 | 1.37±.37 | .022±.011 |
| | | | 1e-3 | Gaussian | ✗ | .81±.75 | 1.06±.28 | .028±.012 |
| | | | 1e-2 | Gaussian | ✗ | .62±.71 | 1.08±.28 | .059±.035 |
| | | | .1 | Gaussian | ✗ | 1.90±4.07 | 1.18±.30 | .039±.016 |
| | | | 1e-5 | Gaussian | ✓ | .64±.71 | 1.04±.26 | .039±.020 |
| Ablations | | | 1e-4 | Gaussian | ✓ | .68±.67 | 1.19±.43 | .057±.033 |
| | | | 1e-3 | Gaussian | ✓ | .75±1.02 | 1.12±.32 | .059±.027 |
| | | | 1e-2 | Gaussian | ✓ | **.47±.48** | **.99±.24** | .069±.040 |
| | | | .1 | Gaussian | ✓ | .89±1.50 | 1.16±.32 | .069±.031 |
| | | | 1e-4 | Plus | ✗ | 1.71±1.15 | 2.46±0.51 | .30±0.062 |
| | | | 1e-2 | Plus | ✓ | 1.16±.61 | 1.85±.26 | .12±.037 |
| | SOM-CPC ($\tau = 0.07$, sim = cosine sim.) | | 1e-4 | Gaussian | ✗ | 1.47±2.60 | 1.15±.34 | **.014±.015** |
| | SOM-CPC ($\tau = 1$, sim = cosine sim.) | | 1e-4 | Gaussian | ✗ | 2.26±4.91 | .96±.13 | .063±.020 |
| | SOM-CPC ($\tau = 0.07$, sim = dot prod.) | | 1e-4 | Gaussian | ✗ | 1.22±4.08 | 1.15±.37 | .066±.046 |
| | CPC + SOM (disjoint) | | - | Gaussian | - | .84±1.14 | 1.47±.50 | .028±.014 |

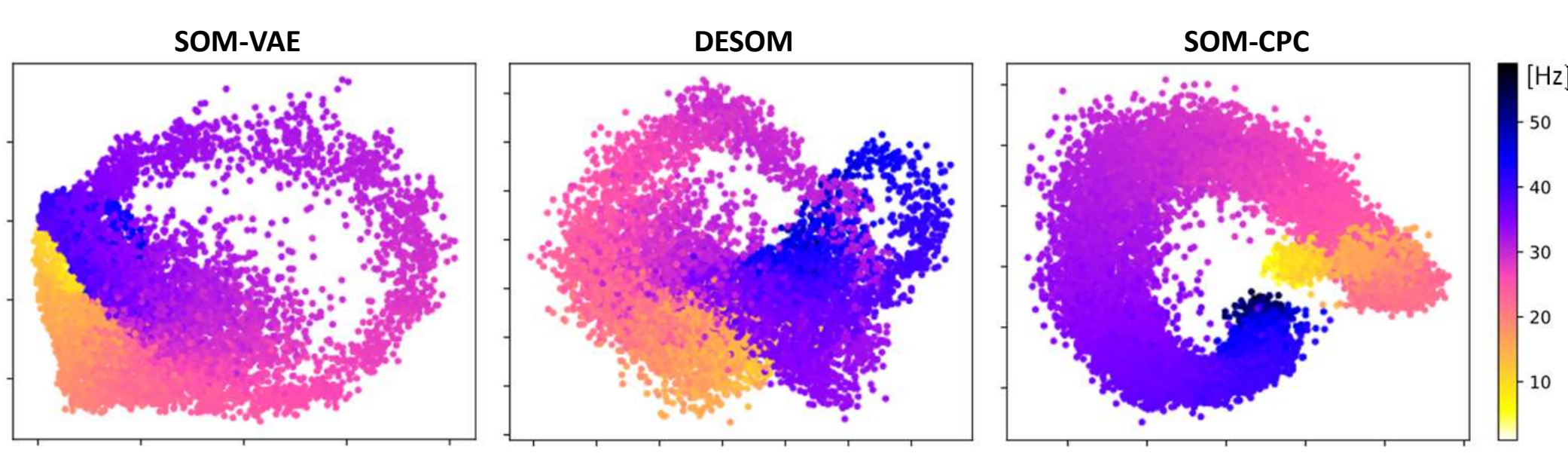

Figure 6: PCA projections of the continuous latent spaces of the full test set for the SOM-VAE, DESOM, and SOM-CPC models of which the SOMs were visualized in fig. 2. Disentanglement of the signal frequencies is much better in the SOM-CPC model.

Figure 6 shows PCA projections of the latent space of the SOM-VAE, DESOM, and SOM-CPC models that are indicated with a * in tables 1 and 3, and for which the SOMs were visualized in fig. 2. Disentanglement of the signal frequencies is much better for the SOM-CPC model, providing an explanation for the better (i.e. lower) $\text{SE}_{\text{target}}$ of this model, as compared to SOM-VAE and DESOM.

### A.4 SLEEP EXPERIMENTS

#### A.4.1 DATA PROCESSING

For the experiments on sleep data, we used subset 3 of the publicly available Montreal Archive of Sleep Studies (MASS) database (O'Reilly et al., 2014), consisting of 62 whole-night polysomnography recordings. Each recording contains, among others, electroencephalography (EEG), chin electromyography (EMG), and electrooculography (EOG) data. We refer the reader to O'Reilly et al. (2014) for more details regarding this dataset. We selected the channels that are typically used in clinical practice, comprising three EEG channels (F4, C4, O2), the two EOG channels, and one chin EMG derivation, and downsampled the data to 128 Hz. Sleep stage labels that follow the guidelines of the American Academy of Sleep Medicine (AASM) Berry et al. (2012) (being wakefulness (Wake), rapid-eye movement (REM) sleep, or non-REM1 till non-REM3 (N1, N2, N3)) were available for every non-overlapping 30-second window, the common label resolution in clinical practice.

Some processing had already been done by the distributors of the MASS dataset (O'Reilly et al., 2014). The 60 Hz powerline interference was, however, not fully suppressed, and we wanted to down sample each signal to 128 Hz to reduce computational complexity. As such, before downsampling, all derivations were additionally filtered with a zero-phase (i.e. two-directional) $5^{\text{th}}$ order Butterworth band-pass filter ($0.3 - 59$ Hz), followed by another zero-phase $5^{\text{th}}$ order Butterworth notch filter ($59 - 61$ Hz). Channels were normalized within-patient and per channel, yielding mean subtraction, followed by normalization such that amplitudes of 95% of the samples were mapped between -1 and +1. The 62 recordings (numbered $1 - 64$, with number 43 and 49 missing) were split into a training set including patients $1 - 48$ ($n = 47$), a validation set including patients $50 - 57$ ($n = 8$), and hold-out test set that included patients $58 - 64$ ($n = 7$).

#### A.4.2 TRAINING DETAILS

Dimensionality reduction of polysomnography data was done by encoding all selected channels in each non-overlapping 30-second window using standard convolutional encoder. Table 4 summarizes the used encoder and decoder (for SOM-VAE and DESOM) architectures. The latent space for decoding in the SOM-VAE and DESOM benchmark models was not fully reduced to a 1D vector to enhance training. Nevertheless, the last adaptive average pooling layer that was used in the encoder of SOM-CPC, was applied in the bottleneck of SOM-VAE and DESOM before SOM quantization took place. As a result the feature vectors in all models were of size $F = 128$. The decoder architecture (see table 4) was used both for the continuous and discrete decoding in the SOM-VAE model (without weight tying). No AR-component was used in the SOM-CPC model to make a fair comparison to the SOM-VAE and DESOM model that also did not include such a component.

For the SOM-CPC model, $P = 3$ future predictions (i.e. positive samples) were used, and $N = 3$ negative samples were drawn for each positive sample. The latter were drawn from the same subject as the positive sample. The $\sigma$ of the Gaussian neighbourhood kernel was exponentially annealed to $\sigma^{(n_{\max})} = 0.5$ during training.

All models were trained with the Adam optimizer (Kingma & Ba, 2014), with a learning rate of 1e-4 and a batch size of 128. Each model was trained for maximally 500 epochs, and the best model was selected based on the lowest $\mathcal{L}_{\text{recon}}$ (for SOM-VAE and DESOM) or $\mathcal{L}_{\text{InfoNCE}}$ (for SOM-CPC and CPC) on the validation set.

#### A.4.3 EXTENDED RESULTS

Table 5 shows the quantitative results on sleep data, comparing SOM-CPC against deep-SOM models (SOM-VAE and DESOM) and disjoint training of CPC, followed by either a supervised linear classifier, K-means or a SOM. Discussion of the main results in this table can be found in section 4.2. The ablation experiments in which the value of $\tau$ and/or the similarity metric was altered show that classification and clustering performance slightly dropped when using the cosine similarity with a temperature value of 1, while the topographic organization slightly improved (i.e. lower TE). These effects vanished when using a temperature of $\tau = 0.07$. Both runs showed worse temporal smoothness (i.e. higher $\ell_{2,\text{smooth}}$). As expected, only changing the temperature value to 0.07 did almost not affect results, suggesting that the linear projector heads were able to adjust for this scaling factor.

Table 4: Model details for the sleep experiments in section 4.2.

| Layer type | Output size | Channels | Activation | Kernel size | Strides | Dilation | Padding |
|---|---|---|---|---|---|---|---|
| **Encoder for SOM-CPC and CPC** | | | | | | | |
| Conv1D | bs $\times 16 \times 128$ | 16 | Leaky ReLU (0.01) | 15 | 1 | 1 | 0 |
| MaxPool1D | bs $\times 16 \times 32$ | - | - | 5 | 5 | - | - |
| Dropout (0.1) | bs $\times 16 \times 32$ | - | - | - | - | - | - |
| Conv1D | bs $\times 32 \times 32$ | 32 | Leaky ReLU (0.01) | 9 | 1 | 1 | 0 |
| MaxPool1D | bs $\times 32 \times 8$ | - | - | 5 | 5 | - | - |
| Dropout (0.1) | bs $\times 32 \times 8$ | - | - | - | - | - | - |
| Conv1D | bs $\times 64 \times 8$ | 64 | Leaky ReLU (0.01) | 5 | 1 | 1 | 0 |
| MaxPool1D | bs $\times 64 \times 2$ | - | - | 5 | 5 | - | - |
| Dropout (0.1) | bs $\times 64 \times 2$ | - | - | - | - | - | - |
| Conv1D | bs $\times 128 \times 2$ | 128 | Leaky ReLU (0.01) | 3 | 1 | 1 | 0 |
| AdaptiveAvgPool1D | bs $\times 128 \times 1$ | - | - | - | - | - | - |
| **Encoder for SOM-VAE and DESOM** | | | | | | | |
| Conv1D | bs $\times 16 \times 128$ | 16 | Leaky ReLU (0.01) | 15 | 1 | 1 | $(18, 17)$ |
| MaxPool1D | bs $\times 16 \times 32$ | - | - | 5 | 5 | - | - |
| Dropout (0.1) | bs $\times 16 \times 32$ | - | - | - | - | - | - |
| Conv1D | bs $\times 32 \times 32$ | 32 | Leaky ReLU (0.01) | 9 | 1 | 1 | 0 |
| MaxPool1D | bs $\times 32 \times 8$ | - | - | 5 | 5 | - | - |
| Dropout (0.1) | bs $\times 32 \times 8$ | - | - | - | - | - | - |
| Conv1D | bs $\times 64 \times 8$ | 64 | Leaky ReLU (0.01) | 5 | 1 | 1 | 0 |
| MaxPool1D | bs $\times 64 \times 2$ | - | - | 5 | 5 | - | - |
| Dropout (0.1) | bs $\times 64 \times 2$ | - | - | - | - | - | - |
| Conv1D | bs $\times 128 \times 2$ | 128 | Leaky ReLU (0.01) | 3 | 1 | 1 | 0 |
| **Decoder for SOM-VAE and DESOM** | | | | | | | |
| Conv1D | bs $\times 64 \times 2$ | 64 | Leaky ReLU (0.01) | 3 | 1 | 1 | 0 |
| ConvTranspose1D | bs $\times 64 \times 2$ | 64 | None | 5 | 5 | 1 | 0 |
| Conv1D | bs $\times 32 \times 2$ | 32 | Leaky ReLU (0.01) | 5 | 1 | 1 | 0 |
| ConvTranspose1D | bs $\times 32 \times 2$ | 32 | None | 5 | 5 | 1 | 0 |
| Conv1D | bs $\times 16 \times 2$ | 16 | Leaky ReLU (0.01) | 9 | 1 | 1 | 0 |
| ConvTranspose1D | bs $\times 16 \times 2$ | 16 | None | 5 | 5 | 1 | 0 |
| Conv1D | bs $\times 6 \times 2$ | 6 | None | 15 | 1 | 1 | 0 |

We also tested the performance when using the SimCLR (Chen et al., 2020) objective for the task loss, instead of the CPC objective. SimCLR is also a contrastive learning framework, but instead of drawing positive samples from the future latent space, these samples are created by applying augmentations on the anchor window. Inspired by Um et al. (2017) we used the following augmentations: independent and identically distributed Gaussian noise $\mathcal{N}(0, 0.05)$ was added (called jitter in their implementation), each channel was scaled with a value drawn from $\mathcal{N}(0, 0.1)$, windows were split in 4 sub-windows of minimal 2 seconds and randomly permuted, and lastly time series were both time warped and magnitude warped. The latter two augmentations make use of smooth curves that smoothly vary the positions of time stamps or magnitude values, respectively.

Besides the difference on how to create positive samples, the originally proposed SimCLR model has some other slight differences with respect to the CPC model:

- The SimCLR loss uses the cosine similarity, while CPC uses the (unnormalized) dot product as the similarity metric (see eq. (9)).

- SimCLR uses an additional temperature $\tau$ in its loss function (see eq. (9)), for which the value is often set to 0.07 (Chen et al., 2020; Woo et al., 2022). CPC does not incorporate such a temperature, which effectively means that it uses a value of 1.

- SimCLR uses a non-linear MLP projection head, while CPC uses linear projection heads.

- SimCLR uses negative samples from within the batch, while this is not specified in the CPC paper. This specified design choice makes SimCLR typically very sensitive to the batch size.

- SimCLR was not proposed to include an auto-regressive component, and can not straightforwardly be extended to do so, while CPC can be implemented with or without such a module.

For the most fair comparison, the procedure for drawing negative samples in SimCLR is done equivalently as for SOM-CPC, i.e. within the recording, instead of within the batch. However, in the SOM-SimCLR model (i.e. the joint training of SOM with SimCLR), each drawn negative sample is added to the set of negative samples both in its raw form, and with a random augmentation, which effectively doubles the number of negative samples. Table 3 reports the performance of the baseline SOM-SimCLR model (i.e. with settings $\tau = 0.07$ and the cosine similarity), and variants using a temperature value of 1 and/or the dot product as the similarity metric. All settings regarding training procedure and the SOM were set equivalently as in the SOM-CPC training. Table 5 shows that SOM-SimCLR results for $\tau = 0.07$ are better than those with $\tau = 1$, which is in line with findings from Chen et al. (2020); Woo et al. (2022). However, even with $\tau = 0.07$, performance of SOM-SimCLR is lower on all metrics compared to the SOM-CPC model with the same value for $\alpha$. The higher $\ell_{2,\text{smooth}}$ metric of SOM-SimCLR indicates on average larger jumps over the SOM

Table 5: Test set performance of various models trained on sleep recordings. SOMs of models with a * are visualized in fig. 4. Bold values indicate the best performance per column (excluding the upper bound of the vanilla CPC model, which does not result in a 2D representation).

| | Model | $\alpha$ | $\mathcal{S}$ | $\mathcal{L}_{\text{SOM}}$ sg[·] | Purity | NMI | Cohen's kappa | $\ell_{2,\text{smooth}}$ | TE |
|---|---|---|---|---|---|---|---|---|---|
| | CPC + linear classifier | - | - | - | - | - | .68±.10 | - | - |
| | CPC + K-means | - | - | - | .79 | .29 | .61±.11 | - | - |
| | CPC ($F=2$) + linear classifier | - | - | - | - | - | .52±.10 | - | - |
| | CPC ($F=2$) + K-means | - | - | - | .74 | .24 | .55±.090 | - | - |
| | CPC + PCA + linear classifier | - | - | - | - | - | .54±.086 | - | - |
| * | CPC + PCA + K-means | - | - | - | .77 | .26 | .57±.082 | - | - |
| | SOM-VAE | 1e-3 | Plus | ✓ | .71 | .23 | .51±.04 | 2.36±.26 | .24±.031 |
| | | 1e-2 | Plus | ✓ | .71 | .23 | .51±.04 | 2.67±.17 | .30±.037 |
| | | .1 | Plus | ✓ | .72 | .23 | .52±.03 | 2.60±.34 | .28±.042 |
| | | 1 | Plus | ✓ | .71 | .23 | .53±.03 | 3.08±.32 | .31±.054 |
| | DESOM | 1e-6 | Gaussian | ✗ | .70 | .27 | .53±.05 | 2.14±.32 | .095±.020 |
| | | 1e-5 | Gaussian | ✗ | .70 | .23 | .50±.04 | 2.10±.26 | .11±.028 |
| | | 1e-4 | Gaussian | ✗ | .71 | .22 | .51±.04 | 2.35±.24 | .17±.035 |
| | | 1e-3 | Gaussian | ✗ | .71 | .22 | .51±.05 | 2.40±.16 | .22±.0085 |
| | | 1e-2 | Gaussian | ✗ | .71 | .22 | .50±.04 | 2.30±.26 | .23±.021 |
| | SOM-SimCLR ($\tau = 0.07$) | 1e-3 | Gaussian | ✗ | .73 | .23 | .53±.13 | 2.21±.35 | .29±.026 |
| | SOM-SimCLR ($\tau = 1$) | 1e-3 | Gaussian | ✗ | .70 | .20 | .48±.16 | 1.87±.30 | .50±.068 |
| | SOM-CPC (ours) | 1e-5 | Gaussian | ✗ | .78 | .27 | .59±.11 | 1.03±.11 | .041±.014 |
| | | 1e-4 | Gaussian | ✗ | .78 | .27 | .61±.10 | **1.01±.10** | .062±.018 |
| * | | 1e-3 | Gaussian | ✗ | .78 | .27 | .61±.12 | 1.02±.09 | .032±.0096 |
| | | 1e-2 | Gaussian | ✗ | **.79** | **.28** | .60±.11 | 1.08±.11 | .19±.042 |
| | | .1 | Gaussian | ✗ | **.79** | **.28** | **.65±.07** | 1.09±.09 | .19±.04 |
| | | 1e-3 | Gaussian | ✓ | .78 | .27 | .62±.10 | 1.02±.12 | .067±.025 |
| | | 1e-2 | Gaussian | ✓ | .78 | .27 | .60±.10 | 1.07±.09 | .17±.04 |
| | | .1 | Gaussian | ✓ | .78 | .27 | .62±.10 | 1.06±.11 | .084±.020 |
| | | 1 | Gaussian | ✓ | .78 | .27 | .59±.12 | 1.04±.12 | .079±.019 |
| | | 1e-3 | Plus | ✗ | **.79** | **.28** | .61±.11 | 1.35±.24 | .26±.079 |
| | | 1e-3 | Plus | ✓ | **.79** | **.28** | .60±.10 | 1.38±.28 | .27±.080 |
| | SOM-CPC ($\tau = 0.07$, sim = cosine sim.) | 1e-3 | Gaussian | ✗ | .78 | .27 | .60±.12 | 1.36±.13 | .070±.023 |
| | SOM-CPC ($\tau = 1$, sim = cosine sim.) | 1e-3 | Gaussian | ✗ | .73 | .27 | .58±.11 | 1.43±.15 | **.025±.0094** |
| | SOM-CPC ($\tau = 0.07$, sim = dot prod.) | 1e-3 | Gaussian | ✗ | **.79** | **.28** | .62±.10 | 1.06±.11 | .059±.020 |
| | CPC + SOM (disjoint) | - | Gaussian | - | **.79** | **.28** | .62±.11 | 1.21±.11 | .52±.042 |

(left margin label: Ablations)

Figure 7: Training curves of SOM-CPC (the model indicated with a * in table 5) for both the training and validation set. The green dashed line indicates the epoch of the used model, i.e. the one with the lowest validation InfoNCE loss. It can be seen that the epoch with the best clustering and classification performance does not necessarily align with the epoch that has the lowest loss commitment and/or SOM loss.

map through time, which might be caused by the fact that the SimCLR task objective does not incorporate temporal information, while InfoNCE does exploit this. Training time of SOM-SimCLR was, moverover, considerably longer than SOM-CPC with the same settings due to the additional augmentations that need to be computed for every data window and its negative samples.

Figure 7 shows training curves of the training and validation set for the SOM-CPC model that is indicated with a * in table 5. The green line indicates the epoch with the lowest InfoNCE validation loss. These graphs show that the performance of InfoNCE, the commitment and SOM loss, and classification metrics do not necessarily align, making it dependent on your final goal with the SOM-CPC model what is the most appropriate stopping-criterion.

## A.5 Audio experiments

### A.5.1 Training details

Audio streams were encoded in windows of 0.01 seconds (=160 samples). For CPC, GRU-DESOM, and SOM-CPC the contextual information of $L = 127$ previous windows was aggregated using a GRU, equivalent as proposed by Oord et al. (2019). The InfoNCE objective for (SOM-)CPC was computed on top of the last context vector of the GRU. To test different settings for the GRU-desom model, we distinguished GRU-DESOM that decodes only the last window, given the last context vector, and GRU-DESOM that decodes the full sequence of 128 windows from the respective context vectors.

Table 6 provides the model details of the encoder, and the decoder for the DESOM benchmarks. The reconstruction loss of the DESOM model was found to be hampered in its optimization when using the encoder architecture, as adopted for the SOM-CPC model. As such, the downsampling factor of the encoder was reduced for the DESOM model to enable minimization of the task loss during training.

Table 6: Model details for the audio experiments in section 4.3.

| Layer type | Output size | Channels | Activation | Kernel size | Strides | Dilation | Padding |
|---|---|---|---|---|---|---|---|
| **Encoder for SOM-CPC and CPC** | | | | | | | |
| Conv1D | bs $\times 512 \times 32$ | 512 | ReLU | 10 | 5 | 1 | 3 |
| Conv1D | bs $\times 512 \times 8$ | 512 | ReLU | 8 | 4 | 1 | 2 |
| Conv1D | bs $\times 512 \times 4$ | 512 | ReLU | 4 | 2 | 1 | 1 |
| Conv1D | bs $\times 512 \times 2$ | 512 | ReLU | 4 | 2 | 1 | 1 |
| Conv1D | bs $\times 512$ | 512 | ReLU | 4 | 2 | 1 | 1 |
| GRU | bs $\times 512$ | 512 | - | - | - | - | - |
| **Encoder for SOM-VAE and DESOM** | | | | | | | |
| Conv1D | bs $\times 512 \times 32$ | 512 | ReLU | 10 | 5 | 1 | 3 |
| Conv1D | bs $\times 512 \times 8$ | 512 | ReLU | 8 | 4 | 1 | 2 |
| Conv1D | bs $\times 512 \times 4$ | 512 | ReLU | 4 | 2 | 1 | 1 |
| Conv1D | bs $\times 512 \times 2$ | 512 | ReLU | 4 | 2 | 1 | 1 |
| Conv1D | bs $\times 512 \times 2$ | 512 | ReLU | 4 | 1 | 1 | same |
| Flatten | bs $\times 1024$ | - | - | - | - | - | - |
| GRU | bs $\times 1024$ | 1024 | - | - | - | - | - |
| **Decoder for SOM-VAE and DESOM** | | | | | | | |
| Unflatten | bs $\times 512 \times 2$ | - | - | - | - | - | - |
| Conv1D | bs $\times 512 \times 2$ | 512 | ReLU | 4 | 1 | 1 | same |
| ConvTranspose1D | bs $\times 512 \times 4$ | 512 | ReLU | 4 | 2 | 1 | 1 |
| ConvTranspose1D | bs $\times 512 \times 8$ | 512 | ReLU | 4 | 2 | 1 | 1 |
| ConvTranspose1D | bs $\times 512 \times 32$ | 512 | ReLU | 8 | 4 | 1 | 2 |
| ConvTranspose1D | bs $\times 1 \times 160$ | 512 | ReLU | 10 | 5 | 1 | 3 (+ output pad = 1) |

For training of SOM-CPC, we followed the settings from Oord et al. (2019) and set $P = 12$. The number of negative samples was set to $N = 10$, which were drawn randomly from the entire training set. All deep-SOM models were trained for maximally 3000 epochs, using the Adam optimizer Kingma & Ba (2014) with a learning rate of 1e-4 and a batch size of 8. One epoch was defined as a push through of one sequence of 128 windows (or 1 window for DESOM) from each recording. The best model was selected based on the lowest validation task loss.

The supervised linear classifier and disjoint SOM training on top of the frozen CPC embeddings were trained with a batch size of 128, and a learning rate of 1e-4 and 1e-2, respectively. Both models were stopped upon convergence of the validation loss (which was after 200 and 250 epochs, respectively).

### A.5.2 Extended results

Quantitative results of the audio experiments can be found in table 7. For this application, Cohen's kappa is computed as the average over all data windows in the test set, which is different from the synthetic and sleep case, where it was computed as the average and one standard deviation across recordings. In these audio experiments, all windows from one recording contain the same speaker id label. Computing Cohen's kappa per-recording, i.e. with having the same label for all windows in that recording, is therefore inappropriate as the computation can not correct for correctness by chance. Table 7 show that SOM-CPC clearly outperformed all variants of the (GRU-)DESOM model, and feature extraction using CPC, followed by PCA and linear or non-linear classification.

Ablations with respect to the temperature $\tau$ and the similarity metric in the loss function indicated, similarly as in the sleep case, that simply changing the temperature value did hardly affect SOM-CPC

performance. However, changing the similarity metric to be the cosine similarity caused a drop in clustering and classification performance, only when using a temperature value of 1. This result overlaps with the experimental findings in the sleep case (see appendix A.4.3), and suggests that a low temperature value - which was found beneficial in the SimCLR objective (Chen et al., 2020; Woo et al., 2022) - does not equivalently improve SOM-CPC performance.

Figure 8 compares the test set projection on the 2D PCA space, created on the CPC features (with $F = 128$), to the SOM from the SOM-CPC model that is denoted with a $*$ in table 7 (and also visualized in fig. 5-right).

Table 7: Test set performance of various models trained on audio recordings. SOMs of models with a * are visualized in fig. 5. Bold values indicate the best performance per column (excluding the upper bound of the vanilla CPC model, which does not result in a 2D representation).

| | Model | $\alpha$ | $\mathcal{S}$ | $\mathcal{L}_{\text{SOM}}$ sg[·] | Purity | NMI | Cohen's kappa | TE |
|---|---|---|---|---|---|---|---|---|
| | CPC + linear classifier | - | - | - | - | - | 1.00 | - |
| | CPC + K-means | - | - | - | 1.00 | .60 | 1.00 | - |
| | CPC ($F = 2$) + linear classifier | - | - | - | - | - | .00 | - |
| | CPC ($F = 2$) + K-means | - | - | - | .13 | .013 | .025 | |
| | CPC + PCA + linear classifier | - | - | - | - | - | .86 | - |
| | CPC + PCA + K-means | - | - | - | .89 | .54 . | .88 | |
| | DESOM | 1e-5 | Gaussian | ✗ | .18 | .03 | .06 | .14±.035 |
| | | 1e-4 | Gaussian | ✗ | .23 | .08 | .11 | .34±.045 |
| | | 1e-3 | Gaussian | ✗ | .31 | .13 | .20 | .68±.050 |
| | | 1e-2 | Gaussian | ✗ | .13 | -.00 | .00 | 1.0±.00 |
| | GRU-DESOM (reconstructing last window) | 1e-5 | Gaussian | ✗ | .19 | .04 | .08 | **.13±.028** |
| | | 1e-4 | Gaussian | ✗ | .26 | .09 | .15 | .20±.036 |
| | | 1e-3 | Gaussian | ✗ | .31 | .13 | .21 | .42±.078 |
| | | 1e-2 | Gaussian | ✗ | .32 | .14 | .22 | .46±.057 |
| | GRU-DESOM (reconstructing full sequence) | 1e-5 | Gaussian | ✗ | .19 | .05 | .08 | .34±.048 |
| | | 1e-4 | Gaussian | ✗ | .30 | .12 | .20 | .59±.072 |
| * | | 1e-3 | Gaussian | ✗ | .33 | .14 | .22 | .78±.048 |
| | | 1e-2 | Gaussian | ✗ | .29 | .12 | .19 | .57±.069 |
| | SOM-CPC (ours) | 1e-5 | Gaussian | ✗ | .99 | **.73** | .99 | .14±.081 |
| | | 1e-4 | Gaussian | ✗ | **1.00** | .63 | **1.00** | .24±.12 |
| * | | 1e-3 | Gaussian | ✗ | **1.00** | .61 | **1.00** | .33±.098 |
| | | 1e-2 | Gaussian | ✗ | **1.00** | .61 | .99 | .33±.099 |
| Ablations | | 1e-3 | Gaussian | ✓ | **1.00** | .61 | .99 | .28±.087 |
| | | 1e-2 | Gaussian | ✓ | **1.00** | .61 | .99 | .35±0.10 |
| | | .1 | Gaussian | ✓ | **1.00** | .61 | **1.00** | .35±.095 |
| | | 1 | Gaussian | ✓ | **1.00** | .61 | **1.00** | .38±.11 |
| | SOM-CPC ($\tau = 0.07$, sim = cosine sim.) | 1e-3 | Gaussian | ✗ | .99 | .61 | .99 | .42±0.12 |
| | SOM-CPC ($\tau = 1$, sim = cosine sim.) | 1e-3 | Gaussian | ✗ | .88 | .55 | .86 | .17±.063 |
| | SOM-CPC ($\tau = 0.07$, sim = dot prod.) | 1e-3 | Gaussian | ✗ | **1.00** | .61 | .99 | .38±.098 |
| | CPC + SOM (disjoint) | - | Gaussian | - | **1.00** | .62 | **1.00** | .28±.11 |

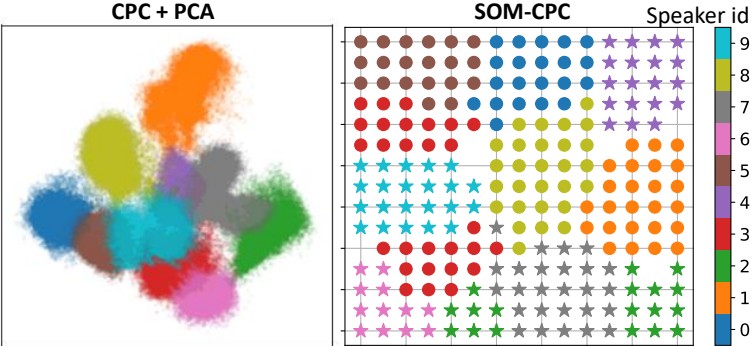

Figure 8: Projecting the test set on the 2D PCA space shows no division of the green and the red clusters in two sub-clusters, something that is visible in the SOM of the SOM-CPC model. These sub-clusters were found to relate to recordings that were made with different room acoustics.

