# OpenReview forum: "SOM-CPC: Unsupervised Contrastive Learning with Self-Organizing Maps for Structured Representations of High-Rate Time Series"
_ICLR.cc/2023/Conference — Submitted to ICLR 2023_

### Official Review · Reviewer_hhoQ · 2022-10-22

**Confidence:** 4
**Correctness:** 3
**Technical Novelty And Significance:** 2
**Empirical Novelty And Significance:** 2
**Recommendation:** 3

**Clarity, Quality, Novelty And Reproducibility:**

**Clarity.** I found the paper reasonably clear although there are minor issues with notation and exposition (see weakness point #4).

**Quality/novelty.** Overall, I find the paper to be very incremental and the experimental results to be insufficient (see weakness points #1, #2, and #3).

**Reproducibility.** This paper appears to be reproducible although I have not done a careful check of the source code that has been provided and various details in the appendices.

**Strength And Weaknesses:**

Strengths:
1. I found the paper fairly straightforward to follow.
2. The proposed method is easy to understand.

Weaknesses:
1. The proposed method is rather incremental. How contrastive learning is incorporated here is really straightforward and a bit obvious.
2. The introduction partially emphasizes interpretability. I would have liked to see much more of a discussion on interpretability in the main paper regarding the real data experiments. Specifically, how does the interpretability of the resulting 2D SOM maps relate to existing domain knowledge for the applications of interest? How does this interpretability compare to those of baseline models that are used (i.e., does the proposed SOM-CPC model somehow yield more interpretable 2D maps than best-performing baselines?)? Moreover, I think that in answering such questions, it would be helpful to be more precise about what you mean by "interpretability" (maybe using terminology from, say, Lipton [2018]).
3. I would argue that more baselines are needed. Right now the baselines are selected to be ones that use 2D embeddings, which seems like an artificially imposed constraint that really isn't required for many real applications (e.g., I'd imagine that with some real high-dimensional datasets, the low-dimensional manifold that the data approximately reside on simply can't be very accurately represented in 2D and could require more dimensions; forcing the use of a 2D embedding in this case seems like a bad idea). There have been a number of temporal clustering methods proposed in recent years, many of which use deep neural networks to help learn representations (such as but not limited to the work by Lee and Van Der Schaar [2020]). Adding some of these methods as baselines would improve the paper as we would have a better sense of how the proposed method compares to other approaches that aren't restricted to 2D embeddings.
4. There are some parts where the notation and exposition are a little clunky and could use more polish. For instance, in the second paragraph of Section 2.1 when notation is first introduced, it is not made immediately clear what the index $j$ is for or what it ranges over, and then suddenly the index $j$ gets switched to $i$ in the next sentence. Minor: The phrase "a given point in training" could also perhaps be more clearly phrased as "a given iteration of the training procedure" where $n$ indexing the iteration number ("a given point" makes it sound like it might be referring to a data point).

References:
- Changhee Lee, Mihaela van der Schaar. Temporal Phenotyping using Deep Predictive Clustering of Disease Progression. ICML 2020.
- Zachary C Lipton. The mythos of model interpretability: In machine learning, the concept of interpretability is both important and slippery. Queue 2018.

**Summary Of The Paper:**

The main technical contribution of this paper is in proposing a variant of deep SOM (self organizing map) model that uses contrastive learning. Experimental results show that the proposed method outperforms various baselines on synthetic and real data.

**Summary Of The Review:**

I think this paper proposes an interesting although fairly straightforward extension to existing deep SOM models that incorporates contrastive learning. More extensive evaluation of "interpretability" and considering additional baselines would really help improve the work although, from what I can tell, neither of these improvements would resolve the issue of the technical novelty being limited.

---

> ### Author Response · Authors · 2022-11-12
> **Part 1**
>
> Weakness 1:
>
> We aimed for a simple, yet very effective model, without additional loss functions or attributes that may be hard to tune, like in the (extensions of the) SOM-VAE model. This makes the SOM-CPC model elegant and easy to use in practice, which we believe is truly a strength, rather than a weakness. We believe the step we made is non-trivial, and potentially a pivot in the deep-SOM literature for three main reasons:
>
> (1) Current trends in deep-SOM literature focus on extensions of the SOM-VAE model, adding auxiliary loss functions and attributes. While these extensions result in improved performance on specific tasks, they make models less elegant, difficult to tune, and therefore not easy to use in practice. We took an entirely different approach, stepping away from the VAE, and replacing deep-SOM's standard reconstruction objective with the InfoNCE objective. We show that this leads to major performance gains while not requiring any auxiliary losses and attributes. We additionally analyzed the impact of each of the elements (neighborhood kernel, gradient blocking, vector-quantization) separately. The result is an easy to use, widely applicable method that sets a new state-of-the-art performance for representation learning of time series.
>
> (2) While CPC on its own has indeed shown merit across a wide variety of tasks, it remained unclear how it would perform in conjunction with the SOM objective. Jointly optimizing these two tasks that are known to have merit separately does not by definition imply that optimizing the joint objective is fruitful as well. As an example, the reconstruction objective, in combination with the SOM loss in the SOM-VAE work showed to work rather poorly on all cases, while autoencoders have been found useful in a plethora of applications.
>
> (3) Our model shows surprisingly strong performance on real-life applications with very little hyperparameter optimization and fine tuning; in particular when compared to current state-of-the-art. This makes it very valuable in our opinion.
>
> Weakness 2:
>
> We thank the reviewer for this interesting remark. We agree with Lipton (2018) that the word interpretability has been used for different meanings. To clarify our definition of interpretability, we have updated our introduction section with the following sentence:
>
> “We define such an interpretable representation as one that is informative, but also facilitates intuitive exploration and knowledge discovery (Lipton, 2018).”
>
> The SOM on sleep data (see Fig. 4), shows a structure that relates very well to domain knowledge about sleep. It is, for example, seen that deep sleep (N3) is positioned far away from lighter sleep (N1) and Wakefulness, with a thick wall of medium-deep sleep (N2) in between. Windows on some nodes in both the Wake and N3 clusters appeared on average earlier in the night (denoted with smaller dots) than windows that were mapped to other nodes, suggesting that there might be different sub-types of Wakefulness and deep sleep N3 that alter throughout the night. We leave such an interpretation and validation of newly-discovered hypotheses to future work.
> In the audio case, the SOM map showed us that different subclusters were present within the red and green speaker, which were found to be caused by different room acoustics in these sub-clusters. The  2D PCA space, applied on the CPC features, would not have given us this insight, since the red and green cluster are not split in this representation. Instead, finding this conclusion without SOM-CPC, would have required listening to all recordings one-by-one. We have added this PCA plot as Figure 8 to Appendix A.5.2, and added the following sentence to Section 4.3:
>
> “This division within recordings of the same speaker is not visible in the 2D PCA projection of the CPC (with F = 128) features, as seen from fig. 8 in appendix A.5.2.”
>
> Given the superior performance of SOM-CPC over the given baselines on various metrics, we can conclude that SOM-CPC is better able to preserve information in its 2D visualization than the baselines.  We hypothesize that marking SOM nodes of the SOM-CPC model with different types of variables facilitates additional knowledge discovery in a wide range of applications. We leave this exploration to future work.

---

> > ### Author Response · Authors · 2022-11-12
> > **Part 2**
> >
> > Weakness 3:
> >
> > We thank the reviewer for this interesting point. The goal of our SOM-CPC model is to facilitate visual, 'user-in-the-loop' pattern recognition in time series data. The reason we chose to set the objective of having a 2D representation is because 2D representations are much more easy to visually interpret than higher-dimensional representations, therewith facilitating scientific discovery, clinical science and diagnoses. Indeed, real-life data can often not accurately be represented in two dimensions. The challenge in mapping to 2D is separating nuisance from important variables. We recognize that this may be application dependent. Nevertheless, we would like to stress that even though a SOM is visualized in 2D, each node’s codebook vector in fact still lives in a higher-dimensional space (as described in section 2.1). This in contrast to, for example, 2D-PCA that simply throws away all information that lives in higher dimensions. The SOM-CPC model thus does create a 2D visualization of time series, without posing the assumption that the data should be representable in 2D, which is truly a strength of this model. We have made this property of SOMs more clear in the introduction of the updated manuscript:
> >
> > “A Self-Organizing Map (Kohonen, 1990), on the other hand, is an extension of K-means clustering that creates a low-dimensional interpretable visualization, while still representing the data in multiple dimensions.”
> >
> > Moreover, we have made explicit in Section 2.1 that we use a 2D representation for the goal of interpretability: “We choose to use a use a 2D visualization to enhance interpretability.”
> >
> > We have compared SOM-CPC to a CPC + K-means baseline that embeds data in F=128 dimensions. SOM-CPC was shown to not lose important information (seen from the very similar performance on all metrics) with respect to CPC + K-means in 128 dimensions. This can be attributed to the power of self-organizing maps of visualizing data in 2D, while preserving more dimensions in the underlying embedding of each node.
> >
> > Our work conceptually differs from the work of Lee et al. (2020) in two ways. First, the goal of Lee et al. (2020) is to cluster a full time series on one variable per time series; expected future outcome. In contrast, our goal is to explore underlying structure (or informativeness, as it is called by Lipton, 2018) within time series by means of interpretable visualizations, which enhances understanding for an expert. Compared to predicting one outcome variable, this per-window visualization allows keeping the expert in the loop while searching for hidden structures, which serves a different goal than outcome prediction.
> >
> > Second, and most important, the method by Lee et al. (2020) uses ground-truth labels to ensure that the found clusters relate to known classes. Our method is fully unsupervised such that it is independent of human biases in labels. A direct comparison on classification and/or clustering metrics seems therefore unfair.
> >
> > We have extended our introduction section with a more thorough discussion on dimensionality reduction and (deep-)clustering methods, including their assumptions, in order to better position our work and make it more clear why we decided to use Self-Organizing Maps with a 2D grid to reach our goal of learning a visually interpretable representation of time series data structure:

---

> > > ### Author Response · Authors · 2022-11-12
> > > **Part 3**
> > >
> > >
> > > “Dimensionality reduction techniques like Principle Component Analysis (PCA), possibly in combination with clustering methods like K-means clustering, have conventionally been used for this purpose. Acquiring an interpretable representation with PCA requires omitting many principle components in order to remain with only few main components. This, however, assumes that the data projects linearly on these low number of dimensions without loss of information. A Self-Organizing Map (Kohonen, 1990), on the other hand, is an extension of K-means clustering that creates a low-dimensional interpretable visualization, while still representing the data in multiple dimensions. However, SOMs typically act on features, which need to be selected heuristically and may, therefore, strongly depend on the use case and/or data modality. Deep learning (DL) models have become popular alternatives for non-linear dimensionality reduction that can be applied directly on raw data streams. Such models have started to be combined with joint clustering objectives in the latent space (Xie et al., 2016; Yang et al., 2017; Madiraju, 2018; Lee & Schaar, 2020). These methods, however, do typically not create a visually interpretable representation, and sometimes make use of label information during training (Lee & Schaar, 2020). To enhance interpretability, latent space representations of DL models are often visualized using a t-distributed stochastic neighbor embedding (t-SNE) (Hinton & Roweis, 2002). Albeit its frequent use, t-SNE does not allow a direct deployment on unseen data as it does not learn a reusable mapping between the multi-dimensional and the low-dimensional space. To acquire visually interpretable data representations from raw data streams, without posing the assumption that data must live in two or three dimensions only, non-linear deep learning encoders have been combined with SOMs (Ferles et al., 2018; Pesteie et al., 2018; Fortuin et al., 2019; Forest et al., 2019; Manduchi et al., 2021; Forest et al., 2021).”
> > >
> > > As an additional baseline we added a comparison to using SimCLR as the task objective, as opposed to CPC, while combining it with jointly training a self-organizing map. Results for the sleep case are shown in the table below. It can be seen that SOM-CPC outperforms SOM-SimCLR, indepdent on temperature value $\tau$ (which was found an important parameter in earlier research to SimCLR). The higher $l_{2,smooth}$ metric indicates on average larger jumps over the SOM map through time, which might be caused by the fact that the SimCLR objective does not incorporate temporal information, while InfoNCE does exploit this.
> > >
> > > $ ~$
> > >
> > > $Model~type~~~~~~Settings~~~~~~~~~~~~~~~~~~Purity~~~~~~NMI~~~~~~Cohens kappa~~~~~~\ell_{2,smooth}~~~~~~~~~~~~~TE$
> > >
> > > SOM-CPC$~~~~~~~~~~~\tau=1~~~~~~~~~~~~~~~~~~~~~~~~~.78~~~~~~~~~~~~~~~.27~~~~~~~~~~~.61\pm.12~~~~~~~~~~~1.02\pm.09~~~~~~~~~~.032\pm.0096$
> > >
> > > SOM-SimCLR$~~~~~\tau=0.07~~~~~~~~~~~~~~~~~~~~.73~~~~~~~~~~~~~~~.23~~~~~~~~~~~.53\pm.13~~~~~~~~~~~2.21\pm.35~~~~~~~~~~.29\pm.026$
> > >
> > > SOM-SimCLR$~~~~~\tau=1~~~~~~~~~~~~~~~~~~~~~~~~~.70~~~~~~~~~~~~~~~.20 ~~~~~~~~~~~.48\pm.16~~~~~~~~~~~1.87\pm.30~~~~~~~~~~.50\pm.068$
> > >
> > >
> > > Weakness 4:
> > >
> > > We thank the reviewer for the detailed observation. We see that the i and j notation can indeed have caused confusion. The i-index is the iterator, while j is a specific assignment of i. We have now updated the notation in Section 2.1 as follows:
> > >
> > > “The $j^{th}$ prototype $\phi_{i=j}^{(n)}=q_{\Phi} (z)$  is the ‘winning vector’ for data point z, at iteration n of the training procedure.”

---

### Official Review · Reviewer_eMcU · 2022-10-24

**Confidence:** 3
**Correctness:** 3
**Technical Novelty And Significance:** 2
**Empirical Novelty And Significance:** 2
**Recommendation:** 6

**Clarity, Quality, Novelty And Reproducibility:**

The paper is well written. The methodology is novel, although the proposed approach is inspired form existing techniques.
The authors provide well structured source code for reproducibility purpose.


**Strength And Weaknesses:**

Strengths:
1.	The paper is well written and clearly organized.
2.	The authors have tested the proposed approach on several applications (real and simulated datasets). Although what is lacking is for example the evaluation of classification performance when using the proposed unsupervised technique (for example in the sleep staging application)
3.	The authors test their approach with state-of-the-art techniques, from which the proposed technique is inspired.
4.	The authors provide well structured source code, which will help greatly for reproducibility of research

Weaknesses:

1.	The authors introduce the concept of aggregate causal context $c(t)$, but explain that the use of this $c(t)$ is not mandatory. The authors do not clearly explain (or I failed to understand it) whether the results in tehri experiment did use the aggregate causal context or not (and did not integrate in the ablation study)
2.	Biomedical time series are by essence non-stationary, therefore the use of contrastive learning may not be optimal for such data. Could the authors expand a bit on this possible drawback?
3.	The explanation of multiple cluster in the audio dataset seem to be a bit overstretched. Explaining that each cluster (for same speaker correspond to a different chapter might be plausible. But did the authors listen carefully to the audio time-series, and did they notice a difference?


**Summary Of The Paper:**

This paper introduces a method for representation learning of multivariate time-series, which consists in the combination of Contrastive learning (for encoding the data in a compact latent space) and Self Organizing Maps (for better interpretability of the latent space). The use of interpretable unsupervised techniques is of the highest importance for biomedical applications, which makes the proposed technique appealing.

**Summary Of The Review:**

This paper introduced an interesting technique for interpretable representation learning, which could be highly interesting for biomedical applications. However biomedical data can be highly non-stationary which might prevent the use of temporal contrastive learning.

---

> ### Author Response · Authors · 2022-11-12
> **Part 1**
>
> Dear eMcU,
>
> We thank you for your time to review our work, and for the positive review. We answer your remaining concerns below.
>
>
> "Although what is lacking is for example the evaluation of classification performance when using the proposed unsupervised technique (for example in the sleep staging application)."
>
> --> Table 5 in Appendix A.4.3 reports the Cohen’s kappa for all models, which is a classification metric (explained in Section 3.3). From all models that reduce the data to a 2D space, SOM-CPC resulted in the highest Cohen’s kappa with respect to the expert annotations (being 0.65).
>
> Weakness 1:
>
> The original CPC model aggregates causal context information in order to predict the future latent space using an auto-regressive component. Adding this component makes the future prediction dependent on a longer history. Without this component, the future prediction is only dependent on the current data window. Thus, the only difference this causal aggregation adds, is the amount of data that is used from the past to predict the future.
>
> In the synthetic use-case, SOM-CPC was found to already work very well without the aggregation of longer contextual information, so it was decided to leave the AR component out for the sake of simplicity and faster training. This also made the direct comparison to the SOM-VAE (and its variants) more fair, since these models do not incorporate an AR module either.
>
> For the sleep use-case, it was seen that the InfoNCE loss overfitted on the training set when incorporating an AR module. As such, it was left out here as well. Also, a difference with respect to the audio use-case, where a GRU was added, is that the window size in the sleep use-case is 30 seconds (which is the clinical standard), while the windows in the audio use-case were 0.01 seconds (as proposed by the original CPC paper). The inclusion of an AR module in the audio-use case did therefore contribute to higher performance, possibly due to the much shorter window length. As such, in the audio model it was added, which facilitated a fair comparison to the GRU-DESOM model.
> We have updated the sentence in Section 3.2 to provide more background on this reasoning:
>
> “Depending on the use case, it was found to not always be necessary, or even beneficial (due to higher risk of overfitting), to use an AR module … "
>
> Weakness 2:
>
> We thank the reviewer for this interesting observation. Indeed, biomedical time series are often non-stationary. This is where the choice for positive and negative samples comes into play. By predicting 3 future windows of 30 seconds in the sleep use case, we assumed that the signal behaved stationary, or at least changed only slowly/in a predictable manner, within a time frame of 120 seconds. Note thus, that the optimal choice for the number of future windows is use-case dependent and important when designing the InfoNCE objective. Nevertheless, in our experience with sleep data we have noticed that the SOM-CPC performance was in practice not very sensitive to the number of future windows. We added some explanation on this topic to the end of our discussion section with the following sentence:
>
> “Moreover, the CPC objective assumes slowly (or non-changing) data characteristics within the time frame in which positive samples are drawn. A multi-modal variational future prediction could possibly improve performance for data that do not meet this assumption.”
>
> Weakness 3:
>
> We listened to the audio recordings and indeed heard a clear acoustic difference between the upper and lower red cluster, and between the left and right green clusters. The chapters belonging to the right green cluster sounded as if they were recorded in a dryer (i.e. less echoic environment) than the recordings in the left green cluster. Similarly, the recordings in the upper red cluster have a fuller ‘body’ than the recordings in the lower red cluster. We have updated the last sentence of Section 4.3 to:
>
> “An auditory inspection revealed that the room acoustics of the recordings belonging to the chapters in different clusters were different, causing changes in the signals which the SOM-CPC model has picked upon.”

---

> > ### Author Response · Authors · 2022-11-12
> > **Part 2**
> >
> > Novelty:
> >
> > We believe the step we made is non-trivial, and potentially a pivot in the deep-SOM literature for three main reasons:
> >
> > (1) Current trends in deep-SOM literature focus on extensions of the SOM-VAE model, adding auxiliary loss functions and attributes. While these extensions result in improved performance on specific tasks, they make models less elegant, difficult to tune, and therefore not easy to use in practice. We took an entirely different approach, stepping away from the VAE, and replacing deep-SOM's standard reconstruction objective with the InfoNCE objective. We show that this leads to major performance gains while not requiring any auxiliary losses and attributes. We additionally analyzed the impact of each of the elements (neighborhood kernel, gradient blocking, vector-quantization) separately. The result is an easy to use, widely applicable method that sets a new state-of-the-art performance for representation learning of time series.
> >
> > (2) While CPC on its own has indeed shown merit across a wide variety of tasks, it remained unclear how it would perform in conjunction with the SOM objective. Jointly optimizing these two tasks that are known to have merit separately does not by definition imply that optimizing the joint objective is fruitful as well. As an example, the reconstruction objective, in combination with the SOM loss in the SOM-VAE work showed to work rather poorly on all cases, while autoencoders have been found useful in a plethora of applications.
> >
> > (3) Our model shows surprisingly strong performance on real-life applications with very little hyperparameter optimization and fine tuning; in particular when compared to current state-of-the-art. This makes it very valuable in our opinion.

---

### Official Review · Reviewer_FiNp · 2022-10-24

**Confidence:** 4
**Correctness:** 4
**Technical Novelty And Significance:** 2
**Empirical Novelty And Significance:** 4
**Recommendation:** 6

**Clarity, Quality, Novelty And Reproducibility:**

Quality: A good paper, well-written. Sound empirical evaluation.
Clarity: Clear, in my opinion.
Originality: Debatable. On one hand, the approach works and no one has tried something similar, as far as I can tell. On the other hand, the main contribution of the paper is applying InfoNCE to a type of problem for which it is intuitively well suited.

**Strength And Weaknesses:**

Strenghts:
- The paper is clear, highlighting the necesary related work to understand the proposed algorithm.
- The empirical evaluation demonstrates good performance across a variety of settings.
- The proposed method is reasonably novel in that it differs from the other instances of SOM-type models.
- Code is released, making reproducing the results more feasible.

Weaknesses:
- The proposed approach reads as a semi-direct application of CPC. CPC (and other objectives similar to InfoNCE) are shown to work across a variety of domains. It is therefore not that surprising that incorporating CPC in a representation task (as one of the two objectives works well).
- I feel a comparative evaluation of different contrastive objectives might be warranted: does adding temperature bring benefits as in [1]? What about MoCo, SimCLR-type objectives (do they work well in this context)?

References:
[1] CoST: Contrastive Learning of Disentangled Seasonal-Trend Representations for Time Series Forecasting


**Summary Of The Paper:**

The submission proposes an algorithm to learn representations of high-rate time series.They focus on a family of models called deep Self Organizing Maps (deep-SOM), models that combine the original SOM objective with (1) neural networks and (2) a task reconstruction loss. The authors introduce a novel variant, SOM-CPC, which follows the deep-SOM formulation, but replaces the task reconstruction loss with an info-NCE objective. Empirical evaluation of their model showcases the strenghts of their proposed approach.


**Summary Of The Review:**

A well-written, clear paper with good empirical contributions. The authors are clearly proposing a working solution to a real problem, hence my rating of at least 6. On the other hand, the proposed approach can be read as marginally derivative, applying an algorithm that intuitively should work for the problem being considered. This in my opinion justifies not giving a higher score.

---

> ### Author Response · Authors · 2022-11-12
> **Part 1**
>
> Dear reviewer FiNp,
>
> We thank you for taking the time to review our work, and for the positive evaluation. Below you can find our replies to the two remaining comments.
>
> Weakness 1:
>
> We believe the step we made is non-trivial, and potentially a pivot in the deep-SOM literature for three main reasons:
>
> (1) Current trends in deep-SOM literature focus on extensions of the SOM-VAE model, adding auxiliary loss functions and attributes. While these extensions result in improved performance on specific tasks, they make models less elegant, difficult to tune, and therefore not easy to use in practice. We took an entirely different approach, stepping away from the VAE, and replacing deep-SOM's standard reconstruction objective with the InfoNCE objective. We show that this leads to major performance gains while not requiring any auxiliary losses and attributes. We additionally analyzed the impact of each of the elements (neighborhood kernel, gradient blocking, vector-quantization) separately. The result is an easy to use, widely applicable method that sets a new state-of-the-art performance for representation learning of time series.
>
> (2) While CPC on its own has indeed shown merit across a wide variety of tasks, it remained unclear how it would perform in conjunction with the SOM objective. Jointly optimizing these two tasks that are known to have merit separately does not by definition imply that optimizing the joint objective is fruitful as well. As an example, the reconstruction objective, in combination with the SOM loss in the SOM-VAE work showed to work rather poorly on all cases, while autoencoders have been found useful in a plethora of applications.
>
> (3) Our model shows surprisingly strong performance on real-life applications with very little hyperparameter optimization and fine tuning; in particular when compared to current state-of-the-art. This makes it very valuable in our opinion.
>
> Weakness 2:
>
> We thank the reviewer for this interesting question and suggestion. Indeed, the temperature parameter, as proposed in SimCLR, and also used in [1], was found to improve performance once it was fixed to a value of 0.07 (as compared to 1 as in the InfoNCE objective). InfoNCE in CPC uses the dot product (i.e. the unnormalized cosine similarit) as similarity metric. Simply changing to another temperature value is hypothesized to not have an effect here, since the linear layers that predict the future latent spaces can simply adjust for this scaling factor.
>
> To test this hypothesis, we have run the SOM-CPC models for all experiments with a temperature of 0.07, both for the dot product and the cosine similarity as similarity metric. Each table below provides the SOM-CPC baseline run (i.e. with temperature 1 and dot product), for the same alpha multiplier as was used for the new runs. For all use cases it can be seen that none of the ablations performed unanimously better nor worse on all metrics. Interestingly, only changing the temperature (i.e. last rows in all tables), indeed showed a very similar performance to the baseline runs (best visible for sleep and audio), suggesting that the linear projector heads indeed adjusted for this scaling factor. Changing the dot product to the cosine similarity, while using a temperature of 1, interestingly, showed lower regression, classification and clustering performance. This lower performance vanished again when using a temperature of 0.07.
> These results suggest that the lower temperature value is mainly beneficial in combination with the cosine similarity, but does not add this much benefit when using the dot product as a similarity metric. We added these results to Tables 3, 5 and 7 in Appendix A.3.2, A.4.3, and A.5.2, respectively, and extended Appendix A.2 with an explanation on this temperature parameter and the similarity metric.
>
> **Synthetic case**:
>
> $~~~~~~~~~~~~~~~~~~~~~~~~~~~~~~~~~~~~~~~~~~~~~~~~~~~MSE ~~~~~~~~~~~~~~~~l_{2,smooth} ~~~~~~~~~~~~~~ TE~$
>
> $\tau=1$, sim=dot product:$~~~~~~~~~~~.72\pm1.08~~~~~~~~~~1.37\pm.37~~~~~~~~~ .022$\pm$.011$
>
> $\tau=0.07$, sim=cosine:$~~~~~~~~~~~~1.47\pm2.60~~~~~~~~~~1.15\pm.34~~~~~~~~~ .014$\pm$.015$
>
> $\tau=1$, sim = cosine:$ ~~~~~~~~~~~~~~~~2.26\pm4.91~~~~~~~~~~~.96\pm.13~~~~~~~~~ .063$\pm$.020$
>
> $\tau=0.07$, sim=dot product:$~~~~1.22\pm4.08~~~~~~~~~~1.15\pm.37~~~~~~~~~ .066$\pm$.046$
>
> $~$
>
> **Sleep case**:
>
> $~~~~~~~~~~~~~~~~~~~~~~~~~~~~~~~~~~~~~~~~~~~~~~~~Purity~~~~~~~~NMI~~~~~~Cohens kappa~~~~~l_{2,smooth} ~~~~~~~~~~~~~~~~ TE~$
>
> $\tau=1$, sim=dot product:$~~~~~~~~~~~~~.78~~~~~~~~~~~~~~~.27~~~~~~~~~~~~ .61\pm.12~~~~~~~~~1.02\pm.09~~~~~~~~.032\pm.0096$
>
> $\tau=0.07$, sim=cosine:$~~~~~~~~~~~~~~~~.78~~~~~~~~~~~~~~~.27~~~~~~~~~~~~.60\pm.12~~~~~~~~~1.36\pm.13~~~~~~~~.070\pm.023$
>
> $\tau=1$, sim = cosine:$ ~~~~~~~~~~~~~~~~~~~.73~~~~~~~~~~~~~~~.27~~~~~~~~~~~~.58\pm.11~~~~~~~~~1.43\pm.15~~~~~~~~.025\pm.0094$
>
> $\tau=0.07$, sim=dot product:$~~~~~~~.79~~~~~~~~~~~~~~~.28~~~~~~~~~~~~.62\pm.10~~~~~~~~~1.06\pm.11~~~~~~~~~.059\pm.020$

---

> > ### Author Response · Authors · 2022-11-12
> > **Part 2**
> >
> >
> > **Audio case**:
> >
> > $~~~~~~~~~~~~~~~~~~~~~~~~~~~~~~~~~~~~~~~~~~~~~~Purity~~~~~NMI~~~~Cohen's kappa~~~~~~~~TE~$
> >
> > $\tau=1$, sim=dot product:$~~~~~~~~~~~1.0~~~~~~~~~~.61~~~~~~~~~~~~ 1.0~~~~~~~~~~~~~~~~~~.33\pm.098$
> >
> > $\tau=0.07$, sim=cosine:$~~~~~~~~~~~~~~.99~~~~~~~~~~.61~~~~~~~~~~~~ .99~~~~~~~~~~~~~~~~~~.42\pm.12$
> >
> > $\tau=1$, sim = cosine:$~~~~~~~~~~~~~~~~~.88~~~~~~~~~~.55~~~~~~~~~~~~ .86~~~~~~~~~~~~~~~~~~.17\pm.063$
> >
> > $\tau=0.07$, sim=dot product:$~~~~~1.0~~~~~~~~~~.61~~~~~~~~~~~~ .99~~~~~~~~~~~~~~~~~~.38\pm.098$
> >
> >
> > Additionally, following the reviewer's suggestion, we have now included a comparison against a SimCLR task loss, jointly trained with the SOM objective. To make the comparison as fair as possible, we trained SimCLR both with a temperature value of 0.07 and 1, and we drew negative samples from within the recording as was also done in the SOM-CPC training. We run tests for the sleep use-case (for which no AR module was used in SOM-CPC), facilitating fair comparison with SimCLR. The procedure and results are added to Appendix A.4.3. Results are also shown in the table below.
> >
> > In line with findings from earlier research, we also see that SOM-SimCLR results for a tau=0.07 are indeed better than those with tau=1.  However, even with tau=0.07, performance of SOM-SimCLR is lower on all metrics compared to SOM-CPC. The higher L2,smooth metric indicates on average larger jumps over the SOM map through time, which might be caused by the fact that the SimCLR objective does not incorporate temporal information, while InfoNCE does exploit this. We, moreover, noticed that training time of SOM-SimCLR was considerably longer than SOM-CPC with the same settings due to the additional augmentations that need to be computed for every data window and its negative samples.
> >
> > $ ~$
> >
> > $Model~type~~~~~~Settings~~~~~~~~~~~~~~~~~~Purity~~~~~~NMI~~~~~~Cohens kappa~~~~~~\ell_{2,smooth}~~~~~~~~~~~~~TE$
> >
> > SOM-CPC$~~~~~~~~~~~\tau=1~~~~~~~~~~~~~~~~~~~~~~~~~.78~~~~~~~~~~~~~~~.27~~~~~~~~~~~.61\pm.12~~~~~~~~~~~1.02\pm.09~~~~~~~~~~.032\pm.0096$
> >
> > SOM-SimCLR$~~~~~\tau=0.07~~~~~~~~~~~~~~~~~~~~.73~~~~~~~~~~~~~~~.23~~~~~~~~~~~.53\pm.13~~~~~~~~~~~2.21\pm.35~~~~~~~~~~.29\pm.026$
> >
> > SOM-SimCLR$~~~~~\tau=1~~~~~~~~~~~~~~~~~~~~~~~~~.70~~~~~~~~~~~~~~~.20 ~~~~~~~~~~~.48\pm.16~~~~~~~~~~~1.87\pm.30~~~~~~~~~~.50\pm.068$

---

### Author Response · Authors · 2022-11-12
**Reply to all reviewers**

Dear reviewers,

We like to thank all of you for taking the time to review our work. We are happy to read that all reviewers found the work well written, and that the work was found “of the highest importance for biomedical applications”, and that it is “clearly proposing a working solution to a real problem”. We cannot agree more that we need representation learning models that also work on real-life data in order to facilitate knowledge distillation.

We have carefully read the feedback of each reviewer and updated the manuscript accordingly to further improve the work. Detailed answers to each reviewer are provided below for each reviewer separately.

---

### Author Response · Authors · 2022-11-17
**What do you think about our responses?**

Dear reviewers,

Some days ago we carefully replied to the raised concerns and updated our manuscript with additional requested experiments.
We believe the work has improved even further, showing the true merit of the SOM-CPC model to do pattern recognition in real-life time series data.

Tomorrow is the last possibility to update the manuscript for us, so could you give us a reply on the additions and comments we made, and re-evaluate your decision?

Thanks in advance.

With kind regards

---

### Decision · Program_Chairs · 2023-01-20

**Decision:**

Reject

**Justification For Why Not Higher Score:**

Technical contributions limited

**Justification For Why Not Lower Score:**

NA

**Metareview: Summary, Strengths And Weaknesses:**

This paper proposes a deep learning method to map time series data to a 2D manifold. The key idea is to jointly optimizes Contrastive Predictive Coding and a SOM objectives. The reviewers find the paper well-written. In the meantime, they also feel that the technical contributions are limited.